# Closed-Form Diffusion Models

**Christopher Scarvelis**                                                    *scarv@mit.edu*
*MIT CSAIL*

**Haitz Sáez de Ocáriz Borde**                                              *chri6704@ox.ac.uk*
*University of Oxford*

**Justin Solomon**                                                          *jsolomon@mit.edu*
*MIT CSAIL*

**Reviewed on OpenReview:** *https://openreview.net/forum?id=JkMifr17wc*

## Abstract

Score-based generative models (SGMs) sample from a target distribution by iteratively transforming noise using the score function of the perturbed target. For any finite training set, this score function can be evaluated in closed form, but the resulting SGM memorizes its training data and does not generate novel samples. In practice, one approximates the score by training a neural network via score-matching. The error in this approximation promotes generalization, but neural SGMs are costly to train and sample, and the effective regularization this error provides is not well-understood theoretically. In this work, we instead explicitly smooth the closed-form score to obtain an SGM that generates novel samples without training. We analyze our model and propose an efficient nearest-neighbor-based estimator of its score function. Using this estimator, our method achieves competitive sampling times while running on consumer-grade CPUs.

## 1 Introduction

Score-based generative models (SGMs) draw samples from a target distribution $\rho_1$ by sampling Gaussian noise and flowing it through a possibly noisy velocity field $v_t$. This velocity field depends on the *score function* of the perturbed target distribution $\rho_t$, which existing SGMs parameterize as a neural network and learn via *score-matching* (Hyvärinen & Dayan, 2005) or denoising (Vincent, 2011; Ho et al., 2020). Although the target distribution $\rho_1$ (for example, the distribution over human face images) is typically assumed to be continuous, in practice score-matching and denoising problems are solved using an empirical approximation $\hat{\rho}_1$ to the target distribution constructed from a finite training set.

When $\hat{\rho}_1$ is the empirical distribution over a finite training set $\{x_i\}_{i=1}^N$, the perturbed target distribution $\rho_t$ is a mixture of Gaussians, whose score function $\nabla \log \rho_t(z)$ has a simple closed-form expression. This score function is a vector pointing from $z$ toward a distance-weighted average of all $N$ rescaled training points and is the *exact* solution to the score-matching problem (Miyasawa, 1961; Raphan & Simoncelli, 2011; Karras et al., 2022). By evaluating this *closed-form score* during sampling, one obtains a training-free sampler for $\hat{\rho}_1$. While this approach seems tempting at first glance, two flaws render it unsuitable for real-world applications:

1. Many applications involve large training sets, prohibiting $O(N)$ computation of the closed-form score.

2. Flowing base samples through the closed-form velocity field simply outputs training samples $x_i$, which is not useful in practice.

Existing work avoids these issues by neurally approximating the score of $\rho_t$. By compressing training data into the score model's weights, neural score functions replace a sum over $N$ training points with a neural network

evaluation whose complexity does not depend directly on $N$. Moreover, neural SGMs generate novel samples given finite training data thanks to approximation error (from limited model capacity) and optimization error (from undertraining) in learning the score (Pidstrigach, 2022; Yoon et al., 2023; Yi et al., 2023). While neural SGMs are successful, they are costly to train, and sampling them requires many (typically GPU-bound) neural network evaluations. Furthermore, the form of the error that enables neural SGMs to generalize is unknown, making it difficult to characterize the distribution from which these models sample in practice.

Our key insight is that the flaws of naïve closed-form SGMs (in particular, lack of generalization and poor scalability) can be addressed without resorting to costly black-box neural approximations. To this end, we make use of a well-known score formula and introduce *smoothed closed-form diffusion models (smoothed CFDMs), a class of training-free diffusion models that require only access to the training set at sampling time.* Smoothed CFDMs generate novel samples from a finite training set by flowing Gaussian noise through a velocity field built from a *smoothed* closed-form score. Our method is efficient, has few hyperparameters, and generates plausible samples in high-dimensional tasks such as image generation. By developing this algorithm, we demonstrate that a closed-form score formula can be adapted to build a non-neural sampler that scales to non-trivial generative tasks.

Our specific contributions are as follows:

1. In Section 4, we show that *smoothing* the exact solution to the score-matching problem promotes generalization by encouraging the score function to point towards barycenters of training samples.

2. Using our smoothed score, in Section 5.1 we construct a *closed-form sampler* that generates novel samples without requiring any training, and characterize the support of its samples.

3. In Section 5.4, we accelerate our sampler using a *nearest-neighbor-based estimator* of our smoothed score, and show in Section 6.2 that in practice, one can aggressively approximate our smoothed score at little cost to sample accuracy.

4. In Section 6, we scale our method to high-dimensional tasks such as image generation. By operating in the latent space of a pretrained autoencoder, we generate novel samples from popular image datasets at speeds competitive with existing GPU-bound methods while running on a *consumer-grade laptop with no dedicated GPU.*

## 2 Related work

*Diffusion models* (Sohl-Dickstein et al., 2015; Song & Ermon, 2019; Ho et al., 2020) have recently achieved state of the art performance in image (Rombach et al., 2022; Zhang & Agrawala, 2023) and video generation (Ho et al., 2022a;b). They have also shown promise in 3D synthesis (Luo & Hu, 2021; Poole et al., 2022; Watson et al., 2022; Lukoianov et al., 2024) and in crucial steps of the drug discovery pipeline such as molecular docking (Corso et al., 2023) and generation (Hoogeboom et al., 2022; Schneuing et al., 2022). Despite this progress, however, diffusion models remain costly to train and sample from (Shih et al., 2023). Prior work has sought to accelerate the sampling of diffusion models via model distillation (Salimans & Ho, 2022), operating in a pre-trained autoencoder's latent space (Vahdat et al., 2021; Rombach et al., 2022), modifying the generative process (Song et al., 2020), using alternative time discretizations for sampling (Zhang & Chen, 2023; Liu et al., 2022; Wu et al., 2023), or by parallelizing sampling steps (Shih et al., 2023). Latent diffusion models also benefit from lower training expenses (Rombach et al., 2022), but publicly-reported training costs for state-of-the-art diffusion models remain high (Bastian, 2022).

Recent works propose alternative diffusion-like models that discard the Markov chain and SDE formalisms from earlier work. Liu et al. (2023) introduce a unified framework for flow-based generative modeling that subsumes diffusion models and show that straightening their model's flows enables few-step sample generation. Heitz et al. (2023) use a similar objective to construct a straightforward graphics-inspired sampler, and Delbracio & Milanfar (2023) concurrently generalize this framework to arbitrary data perturbations and apply it to image restoration and generation tasks. All of these methods parametrize their flows by neural networks that require extensive training.

While diffusion models draw inspiration from mathematical theory (Feller, 1949; Stroock & Varadhan, 1969a;b; 1972), there have been limited attempts to develop a theoretical understanding of their behavior. Salmona et al. (2022); Koehler et al. (2023) study the statistical limitations of diffusion models trained via score-matching, De Bortoli et al. (2021); Lee et al. (2023) present convergence results for diffusion models with absolutely continuous targets, and De Bortoli (2022) extends these results to manifold-supported distributions. However, as diffusion models are trained on an empirical approximation to their target distributions, these results can only show that a diffusion model converges to the empirical distribution of its training set, whereas one is typically interested in generating *novel* samples.

Pidstrigach (2022) takes an initial step in this direction by studying the support of an SGM's model distribution and providing conditions under which an SGM memorizes its training data or learns to sample from the true data manifold. Oko et al. (2023) further show that diffusion models can attain nearly minimax estimation rates for the true data distribution provided its density lies in an appropriate function class. Yoon et al. (2023) propose and empirically test a memorization-generalization dichotomy, which states that diffusion models may only generalize when they are parametrized by neural networks with insufficient capacity relative to the size of their training set. Yi et al. (2023) note that standard training objectives for diffusion models have closed-form optima given finite training sets and show via experiments that the approximation error of neural score functions enables existing diffusion models to generalize. Recently, Kadkhodaie et al. (2024) study generalization in diffusion models using techniques from applied harmonic analysis and demonstrate that SGMs trained on sufficiently large datasets learn a distribution that is effectively independent of the training set, and Aithal et al. (2024) show that neural SGMs "hallucinate" by generating data that lies outside the support of the target distribution because they learn smooth approximations to the ground truth score function. Whereas these works study the generalization of existing SGMs, we construct a novel SGM that explicitly perturbs the closed-form score to generalize without the indeterminate approximation error and training costs of a neural score.

Recent works in graphics and vision have also noted that neural networks are unnecessary for tasks such as novel view synthesis, where neural radiance fields (NeRFs) had previously achieved SOTA results (Barron et al., 2022). In light of this, Kerbl et al. (2023) use efficiently optimized 3D Gaussian scene representations to achieve SOTA visual quality in novel view synthesis while operating in real time. In this work, we adopt a similar perspective and investigate the extent to which neural networks can be replaced with efficient and well-understood classical approaches in generative modeling.

## 3 Preliminaries: The closed-form score

*Flow-based generative models* draw samples from a target distribution $\rho_1$ by sampling from a known base distribution $\rho_0$ (typically $\mathcal{N}(0, I)$) and flowing these samples through a velocity field $v_t$ from $t = 0$ to $t = 1$. For an appropriately-chosen $v_t$, the samples will be distributed according to the target distribution $\rho_1$ at $t = 1$. SGMs employ a $v_t$ that depends on the score function $\nabla \log \rho_t$ of the perturbed data distribution $\rho_t$. For example, when $\rho_0 = \mathcal{N}(0, I)$, one velocity field satisfying this property is $v_t^*(z) = \frac{1}{t}\left(z + (1-t)\nabla \log \rho_t^*(z)\right)$ (Liu et al., 2023), where $\rho_t^*$ is the marginal distribution of the random variable $z = tx + (1-t)\epsilon$, whose samples are target samples $x \sim \rho_1$ that have been rescaled by $t$ and perturbed by Gaussian noise $(1-t)\epsilon \sim \mathcal{N}(0, (1-t)^2 I)$. The score function $\nabla \log \rho_t^*(z)$ is typically learned via score-matching or denoising.

In practice, one learns an SGM from a finite training set $\{x_i\}_{i=1}^N$. In this case, the target distribution $\hat{\rho}_1$ is the empirical distribution over $\{x_i\}_{i=1}^N$, and for the field $v_t^*$ defined above, the perturbed target distribution $\rho_t^*$ is a mixture of Gaussians with means $tx_i$ and common covariance matrix $(1-t)^2 I$. (We will subsequently use the fact that $\rho_t^*$ is a mixture of Gaussians to accelerate our sampler in Sections 5.2 and 6.2.) Its score $\nabla \log \rho_t^*(z)$ has a closed-form expression:

$$\nabla \log \rho_t^*(z) = \frac{1}{(1-t)^2}\left(k_t(z) - z\right), \tag{1}$$

$$\text{where } k_t(z) = \sum_{i=1}^N \text{softmax}\left(-\frac{\|z - tX\|^2}{2(1-t)^2}\right)_i tx_i, \tag{2}$$

in which we let $\|z - tX\|^2$ denote the vector whose $i$-th entry is $\|z - tx_i\|^2$. This $\nabla \log \rho_t^*(z)$ is a vector pointing from $z$ toward a distance-weighted average $k_t(z)$ of all $N$ rescaled training points and is the *exact* solution to the score-matching problem given finite training data. Equation 1 is well-known, having appeared in the empirical Bayes literature as early as in Miyasawa (1961) and more recently in works such as Raphan & Simoncelli (2011) and Karras et al. (2022, Appendix B.3). It has also inspired machine learning methods such as denoising score-matching (Vincent, 2011) and score interpolation (Dieleman et al., 2022).

We define a *closed-form diffusion model* (CFDM) to be the SGM that flows $\mathcal{N}(0, I)$ base samples through this $v_t^*(z)$ while evaluating the score $\nabla \log \rho_t^*(z)$ in closed form as needed during sampling. As this model can only generate samples from the empirical distribution over training data, CFDMs are not useful in practice.

## 4 Smoothed closed-form diffusion models

Pidstrigach (2022); Yi et al. (2023) find that existing diffusion models generalize due to approximation error incurred during score-matching. Rather than studying the generalization of neural SGMs, we take inspiration from this observation and *construct* a training-free SGM that generalizes by explicitly inducing error in the closed-form score.

### 4.1 Definition

Deep neural networks fit the low-frequency components of their target functions first during training, a phenomenon known as "spectral bias" that results in excessively smooth approximations to the target function (Rahaman et al., 2019). Hence, to model the bias of a neural SGM, we induce error in the score function by *smoothing* it. To smooth a function $f$, one chooses a zero-mean noise distribution $p_\epsilon$ and replaces $f(z)$ with the convolution $\tilde{f}(z) = \mathbb{E}_{\epsilon \sim p_\epsilon} [f(z + \epsilon)]$. In practice, we compute the smoothed score function $s_{\sigma,t}(z)$ by fixing a smoothing parameter $\sigma \geq 0$, drawing $M$ noise samples $\epsilon_m \sim p_\epsilon$, and computing

$$s_{\sigma,t}(z) = \frac{1}{(1-t)^2} \left( \frac{1}{M} \sum_{m=1}^{M} k_t(z + \sigma\epsilon_m) - z \right). \tag{3}$$

That is, we *average* the weights $k_t$ in Equation 2 over $M$ small perturbations $\sigma\epsilon_m$ of $z$; as $\sigma \to 0$, we approach the unsmoothed score in Equation 1. We do not add noise to the $-z$ term in the score because it vanishes in expectation. The smoothing procedure in Equation 3 is the key ingredient enabling our model to generalize without a learned approximation to the score function. The smoothed score $s_{\sigma,t}$ can then be inserted into an SGM sampling loop to yield a closed-form sampler that generates novel samples.

To develop intuition for why this simple modification of the closed-form score promotes generalization, we consider the behavior of the closed-form score as $t \to 1$. Figure 1 depicts the closed-form score (Equation 1) and its smoothed counterpart (Equation 3) at $t = 0.95$ for a simple case where the training data consists of two points $x_0$ (in blue) and $x_1$ (in red). In this regime, the temperature $(1-t)^2$ of the softmax in Equation 2

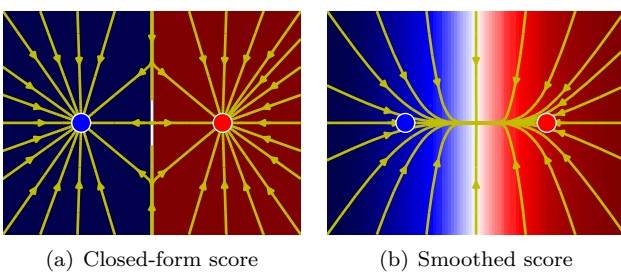

(a) Closed-form score        (b) Smoothed score

Figure 1: Effect of smoothing on the closed-form score (yellow streamplot). Colors represent distance weights in $k_t(z)$; blue regions of space are drawn to the blue point on the left, and vice-versa.

is low, and $k_t(z)$ is effectively equal to the nearest neighbor of $z$ within the training set. Flowing points $z$ through a velocity field such as Liu et al. (2023)'s $v_t^*(z) = \frac{1}{t}(z + (1-t)\nabla \log \rho_t^*(z))$ causes them to flow towards their nearest training sample. As a result, an SGM based on this score function simply outputs training data.

On the other hand, the small perturbations $\sigma\epsilon_m$ in Equation 3 occasionally push points $z$ near the Voronoi boundary between $x_0$ and $x_1$ into their neighboring Voronoi cell. Averaging $k_t$ over these perturbations yields a score function that instead points towards the line segment connecting $x_0$ and $x_1$. An SGM based on the *smoothed* score function will hence cause samples to flow towards weighted barycenters of the training points, which promotes generalization, especially when the data lie on a manifold of sufficiently low curvature. We will make these intuitions rigorous in the following section by proving Proposition 4.1, which will enable us to constrain the support of our model's samples.

### 4.2 Effect of smoothing the score

In this section, we show that as $t \to 1$, the smoothed score points towards barycenters of these tuples rather than towards training points, thereby enabling our sampler to generalize. We first note that via a straightforward computation,

$$k_t(z + \sigma\epsilon_m) = \sum_{i=1}^N \mathrm{softmax}\left(-\frac{\|z - tX\|^2 + \sigma t u_{i,m}}{2(1-t)^2}\right)_i tx_i, \tag{4}$$

where $u_{i,m} = -2\langle\epsilon_m, x_i\rangle$ is a scalar random variable. This shows that smoothing the score acts by perturbing the distance weights $-\|z - tx_i\|$, so one can directly add scalar noise $u_{i,m} \sim p_u$ to these weights instead of perturbing the inputs $z$ with noise $\epsilon_m \sim p_\epsilon$. To simplify our exposition, we will frame the remainder of our results from this perspective.

We now show that smoothing the closed-form score yields a function $s_{\sigma,t}(z)$ that points from $z$ towards a convex combination $k_{\sigma,t}(z)$ of *barycenters* $t\bar{c}_k = \frac{1}{M}\sum_{m=1}^M tx_{i(k,m)}$ of tuples $tC_k = (tx_{i(k,m)} : m = 1, ..., M)$ of rescaled training points. In this notation, $i(k,m)$ picks out an index $i$ corresponding to one of the $N$ training points $\{x_1, ..., x_N\}$ that depends both on the identity of the tuple $tC_k$ (represented by the argument $k$ and the index's position in the tuple (represented by the argument $m$). In this way, the $k$-th tuple $tC_k$ contains $M$ training samples $x_{i(k,m)}$ for $m = 1, ..., M$. The weights of this convex combination depend not only on the distance $\|z - t\bar{c}_k\|$ between $z$ and the barycenters $t\bar{c}_k$, but also on the *variance* of the tuples $tC_k$ and the noise terms $\bar{u}_k = \frac{1}{M}\sum_{m=1}^M u_{i(k,m)}$. Tuples of tightly-clustered points have low variance and hence receive large weights in $k_{\sigma,t}(z)$, whereas tuples of distant points have high variance and receive small weights in $k_{\sigma,t}(z)$. We prove the following proposition in Appendix B.1.

**Proposition 4.1** ($s_{\sigma,t}$ *points towards barycenters of training points*)**.** *The smoothed score $s_{\sigma,t}(z)$ can be expressed as:*

$$s_{\sigma,t}(z) = \frac{1}{(1-t)^2}\left(k_{\sigma,t}(z) - z\right), \ \ where$$

$$k_{\sigma,t}(z) = \sum_{k=1}^{N^M} softmax\left(-\frac{M\left(\|z - t\bar{c}_k\|^2 + Var(tC_k) + \sigma t\bar{u}_k\right)}{2(1-t)^2}\right)_k t\bar{c}_k. \tag{5}$$

## 5 Sampling algorithm

### 5.1 Forward Euler scheme for sampling

Armed with the smoothed score $s_{\sigma,t}$, we are now in position to define our sampler. Following Liu et al. (2023), we draw $\mathcal{N}(0, I)$ base samples and flow them through

$$v_{\sigma,t}(z) = \frac{1}{t}\left(z + (1-t)s_{\sigma,t}(z)\right), \tag{6}$$

---

**Algorithm 1** Sampling

---

**Input:** Training set $\{x_i\}_{i=1}^N$, noise $\{u_{i,m}\}$, step size $h = \frac{1}{S}$, initial sample $z_0 \sim \mathcal{N}(0, I)$
    **for** $n = 0, ..., S - 1$ **do**
        $t_n = \frac{n}{S}$
        $z_{n+1} \leftarrow z_n + hv_{\sigma,t_n}(z_n)$
    **end for**
    **return** $z_T$

---

from $t = 0$ to $t = 1$. We discretize this ODE using a forward Euler scheme, leading to Algorithm 1 for sampling using the smoothed score.

The smoothed score in Equation 3 and Algorithm 1 jointly define our *smoothed closed-form diffusion model*; given a smoothing parameter $\sigma$, we call this a $\sigma$-CFDM. Using Algorithm 1, we can sample from a $\sigma$-CFDM given access only to the training data $\{x_i\}_{i=1}^N$ and noise samples. Notably, no training phase or neural network is needed for this procedure. By explicitly smoothing the closed-form score rather than relying on a neural network's approximation error, we can determine the support of our $\sigma$-CFDM's distribution. For sufficiently small step sizes, our model's samples will lie at the barycenters of tuples of training points.

**Theorem 5.1** (Support of $\sigma$-CFDM samples). *All samples returned by Algorithm 1 are of form $z_S = \frac{S}{S-1} k_{\sigma, \frac{S-1}{S}}(z_{S-1})$. As the number of sampling steps $S \to \infty$ (equivalently, as the step size $\frac{1}{S} \to 0$), the model samples converge towards barycenters $z_S = \bar{c}_k$ of $M$-tuples of training points.*

We prove this theorem in Appendix B.2. While our sampler is easy to implement and training-free, it may be costly if the number of training samples $N$ and the number of sampling steps $S$ are large. We address these issues in the following sections. In Section 5.4, we show how to approximate our smoothed score using efficient nearest-neighbor search. In Section 6.2, we demonstrate that one may take fewer sampling steps by initializing the sampler at a non-zero start time at little cost to sample accuracy, and provide complementary analysis in Section 5.2. This will permit our method to scale to high-dimensional real-world datasets while achieving sampling times competitive with existing methods and running on consumer-grade CPUs.

## 5.2 Taking fewer sampling steps

As a CFDM's distribution $\rho_t^*$ is simply a time-dependent mixture of Gaussians centered at the training points, it can be directly sampled at any time $t$ by uniformly sampling a mixture mean $tx_i$ and then sampling from a Gaussian centered at $tx_i$. We use this fact to sample a $\sigma$-CFDM with fewer steps by starting at $T > 0$ with samples from its corresponding unsmoothed CFDM. As a $\sigma$-CFDM does not have the same distribution as an unsmoothed CFDM, this approximation incurs some error, which we bound in the following theorem.

**Theorem 5.2** (Approximation error from starting at $T > 0$). *Let $\rho_{1-\epsilon}^T$ be the model distribution at $t = 1 - \epsilon$ obtained by starting sampling a $\sigma$-CFDM at $T > 0$ with samples from the unsmoothed CFDM, and let $\rho_{\sigma,1-\epsilon}^0$ be the corresponding $\sigma$-CFDM model distribution when sampling starting at $T = 0$. Then for any fixed $T$ and $\epsilon$,*

$$W_2(\rho_{\sigma,1-\epsilon}^0, \rho_{\sigma,1-\epsilon}^T) = O(\sigma). \tag{7}$$

*where $W_2$ is the 2-Wasserstein distance.*

Following De Bortoli (2022), we stop sampling at time $1 - \epsilon$ for some truncation parameter $\epsilon > 0$ to account for the fact that the smoothed score $s_{\sigma,t}$ blows up as $t \to 1$ due to division by $(1 - t)^2$. We prove this theorem in Appendix B.3.

This result shows that initializing a $\sigma$-CFDM with samples from the unsmoothed CFDM $\rho_T^*$ at time $T > 0$ results in bounded error that scales linearly with $\sigma$. Intuitively, increasing $\sigma$ causes the unsmoothed velocity field $v_t^*$ to be a worse approximation to the smoothed velocity field $v_{\sigma,t}$ at any time $t$; Theorem 5.2 confirms that the cost to sample accuracy is linear in $\sigma$.

### 5.3 Distribution of one-step samples under Gumbel weight perturbations

When the scalar noise $u_{i,m}$ perturbing the distance weights in Equation 4 is drawn from a Gumbel$(0,1)$ distribution, we can precisely characterize the smoothed model's distribution when performing *single-step sampling* by starting sampling at the final Euler iteration in Algorithm 1.

**Proposition 5.3.** *Suppose we begin sampling a smoothed CFDM at iteration $S-1$ of Algorithm 1 using samples $z_{S-1} \sim \rho^*_{t_{S-1}}$ from the unsmoothed CFDM at $t_{S-1}$. Suppose also that the perturbations $u_{i,m}$ to the distance weights in Equation 4 are drawn from a Gumbel$(0,1)$ distribution. Then, as the number of Euler steps $S \to \infty$, the model samples $z_S$ are of the form $z_S = \frac{1}{M} X I_\sigma$, where $X \in \mathbb{R}^{D \times N}$ is the matrix whose i-th column is training sample $x_i \in \mathbb{R}^D$ and $I_\sigma \sim Multinomial(\pi_\sigma, M)$. The probability $\pi^i_\sigma$ of training point $x_i$ is given by $\pi^i_\sigma = softmax\left(-\frac{1}{\sigma}\|z_{S-1} - x_i\|^2\right)$.*

We prove this proposition in Appendix B.4. This result provides further intuition on the role of the smoothing parameter $\sigma$ in determining the distribution of a smoothed CFDM's samples: It is the *temperature* of the softmax determining $\pi^i_\sigma = \text{softmax}\left(-\frac{1}{\sigma}\|z_{S-1} - x_i\|^2\right)$. When $\sigma = 0$, the softmax simply picks out the training sample $x_i$ that is closest to $z_{S-1}$. Conversely, as $\sigma \to \infty$, the event probabilities $\pi^i_\sigma$ become uniform and $z_S$ becomes the barycenter of $M$ uniformly-sampled training points.

### 5.4 Fast score computation via approximate nearest-neighbor search

Each sampling step in Algorithm 1 requires the evaluation of the smoothed score $s_{\sigma,t}(z)$ and hence a sum over $O(N)$ terms. For large datasets, each evaluation of $s_{\sigma,t}(z)$ is therefore costly and places substantial demands on memory.

In the $t \to 1$ regime, the temperature of the softmax in Equation 4 is low, and the large sum is dominated by the handful of terms corresponding to the smallest values of $\|z - tx_i\|^2 - \sigma t u_{i,m}$. If $\sigma$ is sufficiently small, these terms correspond to the *nearest neighbors* of $z$ among the rescaled training points $tx_i$. This suggests that we can approximate the smoothed score $s_{\sigma,t}$ by subsampling terms in the $O(N)$ sum while ensuring that the nearest neighbors of $z$ are included with high probability.

Noting that the closed-form score $\nabla \log \rho^*_t(z) = \frac{\nabla \rho^*_t(z)}{\rho^*_t(z)}$ is the score of a Gaussian kernel density estimator (KDE) $\rho^*_t$, we employ Karppa et al. (2022)'s unbiased nearest-neighbor estimator for KDEs to estimate the denominator, and take its gradient to obtain an unbiased estimate of the numerator. We provide details on this estimator in Appendix A. Our estimator is computed using the $K$ nearest neighbors of $z$ in the training set and $L$ random samples from the remainder of the training set; we study the accuracy tradeoffs associated with $K$ and $L$ in Section 6.2.

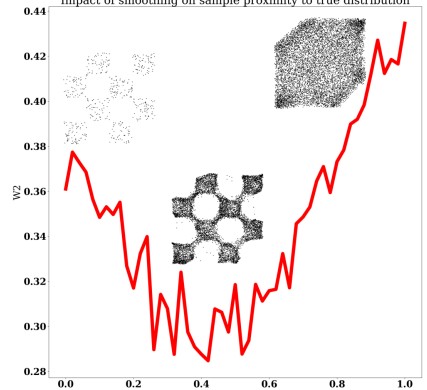

Given this estimator for the closed-form score, we estimate the smoothed score $\widehat{s_{\sigma,t}}$ via convolution against a smoothing kernel as in Section 4.1. By using the approximate nearest neighbor search algorithms implemented in Faiss (Douze et al., 2024), we are able to scale our method to high-dimensional real-world datasets and achieve sampling times competitive with neural SGMs while running on consumer-grade CPUs; see Sections 6.3 and 6.4 for examples and runtime metrics.

## 6 Results

### 6.1 Impact of $\sigma$ on generalization

We now show that a $\sigma$-CFDM's model distribution approaches the true distribution $\rho_1$ of its training samples $x_i \sim \rho_1$ for appropriate values of $\sigma$. We fix a continuous target distribution $\rho_1$ and draw $N = 5000$ samples $y_i$ to serve as a discrete approximation to $\rho_1$. We then draw a smaller subset of $n = 500$ training samples $x_i$ and construct a $\sigma$-CFDM on these samples while varying $\sigma$.

Figure 2: $W_2$ between $\sigma$-CFDM model samples and true samples. We depict model samples for $\sigma \in \{0, 0.26, 1\}$.

For each $\sigma$, we measure the 2-Wasserstein distance $W_2$ between the $\sigma$-CFDM's generated samples and the true samples $y_i \sim \rho_1$, and use this as a tractable proxy for the distance between the $\sigma$-CFDM's model distribution and the true distribution $\rho_1$. We present the results of this experiment for the "Checkerboard" distribution in Figure 2.

When $\sigma = 0$, the support of our model's samples (left side of Figure 2) coincides with the training samples $x_i$. The 2-Wasserstein distance between the model samples and true samples $y_i$ decreases for small values of $\sigma$ as the model samples become convex combinations of nearby points in the training set; we depict model samples for $\sigma = 0.26$ in the center of Figure 2. However, as $\sigma$ grows large, the model samples spread out to fill the convex hull of the training set (right side of Figure 2) and the distance between our model's samples and true samples $y_i$ grows rapidly. These results suggest that for appropriate values of $\sigma$, our method can use a fixed training set $\{x_i\}_{i=1}^N$ to generate novel samples that remain close to the target distribution $\rho_1$. Experiments of this type may be used to select appropriate values of $\sigma$ for a given application.

In Figure 3, we demonstrate that with an appropriate choice of $\sigma$, our method can sample from a 2D surface embedded in $\mathbb{R}^3$ given a sparse blue noise sampling of the surface; this is a low-dimensional case of a manifold-supported distribution, which is typical in machine learning applications. Our method's samples (blue points) fill in the gaps between the sparse training samples (red points) while remaining close to the true manifold. This occurs because $\sigma$-CFDM samples are barycenters of tuples of nearby training points, with $\sigma$ controlling the variance of these tuples. For appropriate values of $\sigma$ and sufficiently dense samplings of training points, these barycenters will approximately lie on tangent planes to the surface, and hence lie near the surface but away from the training data.

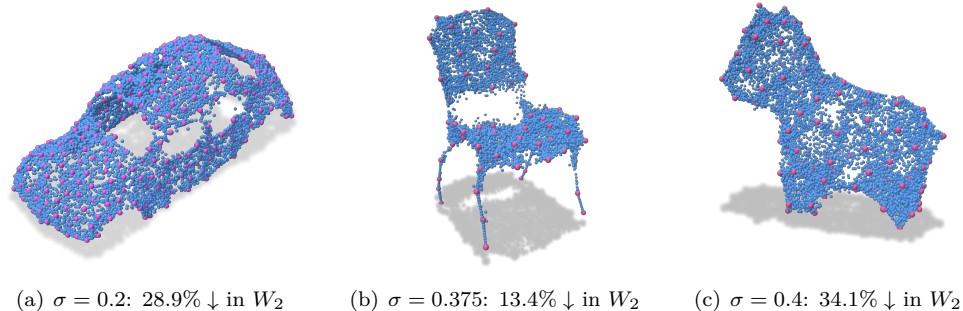

    (a) $\sigma = 0.2$: 28.9% ↓ in $W_2$     (b) $\sigma = 0.375$: 13.4% ↓ in $W_2$     (c) $\sigma = 0.4$: 34.1% ↓ in $W_2$

Figure 3: Sampling a $\sigma$-CFDM (blue points) yields a dense point cloud given sparse mesh samples (red points). We report % drop in $W_2$ distance to a dense mesh sampling when using our $\sigma$-CFDM's samples relative to the sparse training samples. We render these point clouds in Polyscope (Sharp et al., 2019).

## 6.2 Ablation and computational trade-offs

In this section, we investigate the impact of the start time $T$ and the parameters of our nearest-neighbor-based score estimator (9) on the distribution of our model's samples.

**Impact of $T$.** As a CFDM's distribution $\rho_t^*$ is simply a time-dependent mixture of Gaussians centered at training points, it can be directly sampled at any time $t$ by uniformly sampling a mixture mean $t x_i$ and then sampling from a Gaussian centered at $t x_i$. We use this fact to sample a $\sigma$-CFDM with fewer steps by starting at $T > 0$ with samples from its corresponding unsmoothed CFDM. We show here that for practical values of $\sigma$, one can begin sampling at $T$ close to 1 with little accuracy loss.

We fix a continuous target distribution $\rho_1$, draw $n = 500$ training samples $x_i$, and construct a $\sigma$-CFDM on these samples for $\sigma \in \{0, 0.2, 0.4, 0.6, 0.8, 1.0\}$. We then vary the initial sampling times $T$ and compute the 2-Wasserstein distance $W_2$ between model samples generated starting at $T = 0$ and at $T > 0$. We compare this to the average $W_2$ distance between batches of $\sigma$-CFDM samples generated by starting at $T = 0$ (which

is nonzero due to randomness in sampling) and report the percent change in $W_2$ relative to this baseline value. We present the results of this experiment for the "Checkerboard" distribution in Figure 4.

For $\sigma < 0.4$, there is little accuracy loss when starting at $T > 0$, even for start times close to 1. When $\sigma \geq 0.4$, the accuracy of this approximation begins to decline for start times $T \geq 0.4$, with large reductions in approximation quality when both $\sigma$ and $T$ are large. As we have found that our model has performed best with $\sigma \leq 0.4$ in the applications considered in this work, this section's results support the use of few sampling steps in practice. The results in Sections 6.3 and 6.4 further support the use of late start times $T$ for image generation; we find in these experiments that we can start sampling as late as $T = 0.98$ while maintaining good sample quality.

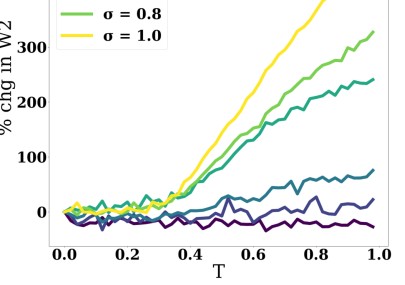

**Impact of $K$ and $L$ on the NN-based score estimator.** In Section 5.4, we proposed an efficient score estimator based on fast nearest-neighbor search. We now study the impact of the number of nearest neighbors $K$ and the number of random samples $L$ from the remainder of the training set on our model's samples.

Figure 4: % change in $W_2$ between $\sigma$-CFDM model samples generated starting at $T = 0$ and samples generated starting at $T > 0$.

We fix a continuous target distribution $\rho_1$, draw $n = 500$ training samples $x_i$, and construct a $\sigma$-CFDM on these samples for $\sigma = 0.3$; this value is typical for real-world applications. We then vary the number of nearest neighbors $K$ and the number of random samples $L$ used to compute the score estimator Equation 9 and measure the 2-Wasserstein distance between model samples generated using the full smoothed score and using the estimator Equation 9. We present the results of this experiment for the "Checkerboard" distribution in Figure 5. We center the diverging color scheme at the $W_2$ distance between two batches of samples from a $\sigma$-CFDM using the full smoothed score; this noise threshold encodes the inherent randomness in our model's samples across batches.

The error arising from the NN-based estimator is decreasing in $K$ and $L$, with especially poor approximation quality when using a single random sample $x_\ell$. However, the accuracy of the model samples approaches the noise threshold for small values of $K, L$. For example, with $K = L = 15$ (which samples just 6% of the terms in $k_{\sigma,t}$), the $W_2$ distance between samples generated using the full score and the NN-based estimator is 0.1865, a value close to the noise threshold of 0.1791. In Sections 6.3 and 6.4, we additionally show that one can generate high-quality images while subsampling $k_{\sigma,t}$ at a far lower rate, thereby enabling our method to scale to real-world datasets.

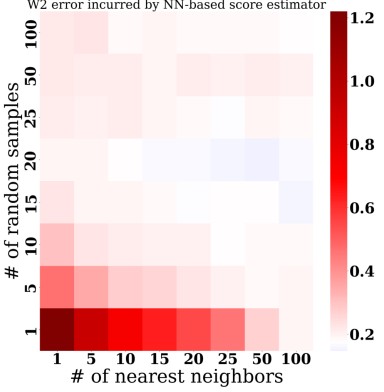

## 6.3 Image generation in pixel space

In this section, we use our $\sigma$-CFDM to sample images in pixel space that are similar to images from the "Smithsonian Butterflies" dataset,[1] rescaled to $128 \times 128$. We benchmark our model's sample quality, training time, and sampling time against a denoising diffusion probabilistic model (DDPM) (Ho et al., 2020) and provide training details in Appendix C.1.

Figure 5: $W_2$ between $\sigma$-CFDM model samples generated using the full score and our NN-based estimator for varying # of NN $K$ (horizontal axis) and # of random samples $L$ (vertical axis).

We display images from a held-out test set along with DDPM samples and our model's samples in Figure 6. Both our model and the DDPM generate images that qualitatively resemble the test images, but as our model can only output barycenters of training samples (see Theorem 5.1), our samples exhibit softer details than the test and DDPM samples. Table 1 records sample quality metrics and training and generation times for our method and the DDPM baseline. Our training-free method achieves

---

[1]Dataset available on Hugging Face: `huggan/smithsonian_butterflies_subset`

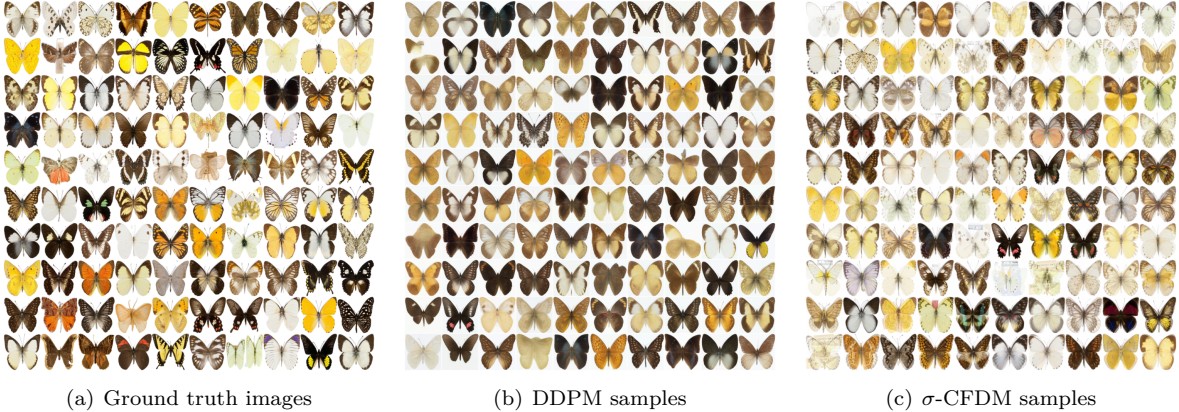

(a) Ground truth images          (b) DDPM samples          (c) $\sigma$-CFDM samples

Figure 6: Ground truth images from Smithsonian Butterflies (left), DDPM samples (center), and $\sigma$-CFDM samples (right).

comparable sample quality to a DDPM that has been trained for 5.34 hours on a single V100 GPU, and achieves over 2.9 times the sample throughput of a DDPM running on a V100 GPU while running on a Macbook M1 Pro CPU with 16 GB of RAM.

As $128 \times 128$ RGB images lie in 49,152-dimensional space, this experiment demonstrates that our method scales to high-dimensional problems. As our method is able to generate plausible samples despite being restricted to outputting barycenters of training samples, it also demonstrates that there exist image manifolds for which our $\sigma$-CFDM's inductive bias is reasonable. However, we do not expect this inductive bias to be suitable for most real-world image data, where barycenters of training samples typically lie off-manifold and fail to resemble ground truth samples. To narrow this gap between theory and practice, we show in the following section that by sampling in the latent space of an autoencoder, our method can generate plausible and diverse images of human faces, comparing favorably with a VAE at marginally higher sampling costs.

Table 1: Metrics for sample quality and generation time in pixel space. Our $\sigma$-CFDM achieves competitive sample quality and generation time while requiring no costly training.

| Method | Metric | Butterflies |
|---|---|---|
| DDPM | Inception score ↑ | $1.87 \pm 0.225$ |
| | KID ↓ | $0.0220 \pm 0.0038$ |
| | Training time | 5.24 h |
| | Sampling time (GPU) | 1.20 s |
| $\sigma$-CFDM | Inception score ↑ | $2.20 \pm 0.150$ |
| | KID ↓ | $0.0114 \pm 0.0048$ |
| | Training time | 0 h |
| | Sampling time (CPU) | 0.4124 s |

## 6.4 Image generation in latent space

Theorem 5.1 shows that in the limit of small step sizes, a $\sigma$-CFDM's samples are barycenters of nearby training points. This is typically a poor prior for images in pixel space, but an appropriately-chosen autoencoder may map the training data to a latent manifold that more closely satisfies this local linearity assumption. To this end, we train the nuclear norm-regularized autoencoder proposed by Scarvelis & Solomon (2024), which encourages latent vectors to lie on a low-dimensional manifold. We then sample from a $\sigma$-CFDM in the latent space of this pretrained autoencoder for the CelebA dataset (Liu et al., 2015) and discard samples that are identical to their nearest latents from the training set. As our method relies on tractable nearest-neighbor queries in the training set at sampling time, this is a feasible post-processing step for our

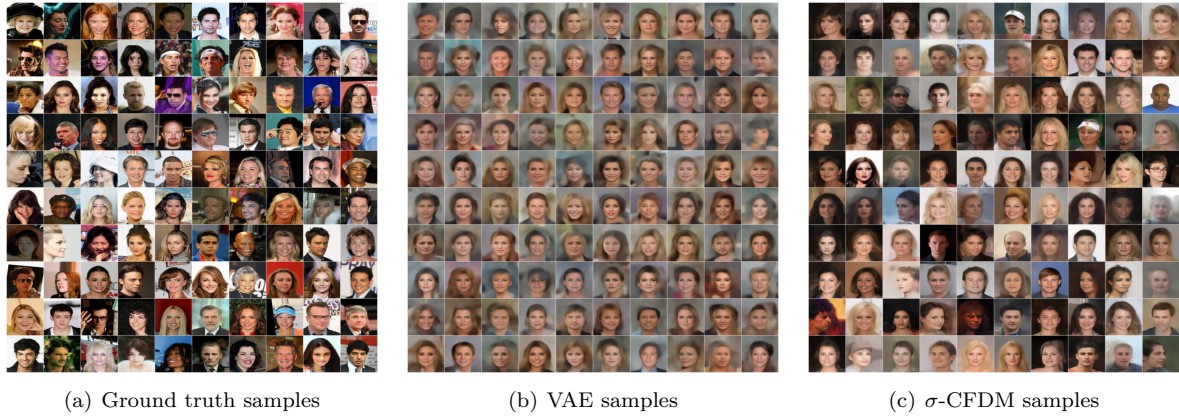

(a) Ground truth samples      (b) VAE samples      (c) $\sigma$-CFDM samples

Figure 7: Ground truth images from CelebA (left), VAE samples (center), and $\sigma$-CFDM samples (right).

sampler. We benchmark our method against a Variational AutoEncoder (VAE) (Kingma, 2013) trained for the same number of epochs and employing the same architecture as the nuclear norm-regularized autoencoder. We train both autoencoders using the log hyperbolic cosine reconstruction loss, which has been found to improve sample quality in VAEs (Chen et al., 2019).

A VAE is an appropriate baseline for a latent $\sigma$-CFDM because both models train a regularized autoencoder to obtain well-structured latent representations, and then employ a training-free process to generate new samples in latent space. A VAE's sampling procedure is simple and data-independent: One draws a normally-distributed latent and decodes it. However, one must use heavy regularization to ensure the latent distribution is nearly Gaussian, and VAEs suffer from poor sample quality as a result. In contrast, a $\sigma$-CFDM's data-dependent sampling procedure merely requires that the latent distribution be supported on a manifold of sufficiently low curvature, so that barycenters of nearby latents continue to lie on this manifold. We consequently expect that one may sample a $\sigma$-CFDM in a weakly regularized latent space to obtain better-quality decoded samples than a VAE while preserving the ability to sample on CPU without requiring additional training.

We display our model's samples, along with VAE samples and ground truth samples from the CelebA dataset in Figure 7. Barycenters of natural images in pixel space typically do not resemble natural images unless they are well-registered (as with the butterflies in Section 6.3), but operating in an autoencoder's latent space allows our method to generate plausible and diverse images of human faces. In particular, our method's samples exhibit greater qualitative diversity than the VAE samples, at times including features such as hats and glasses that seldom or never appear in the VAE baseline's samples.

Table 2: Metrics for sample quality and generation time in latent space. Our $\sigma$-CFDM improves significantly on a VAE's sample quality at a marginal sampling cost on CPU.

| Method | Metric | CelebA |
|---|---|---|
| | Inception score $\uparrow$ | $1.68 \pm 0.08$ |
| VAE | KID $\downarrow$ | $0.108 \pm 0.0066$ |
| | Sampling time (CPU) | $-$ |
| | Inception score $\uparrow$ | $2.22 \pm 0.19$ |
| $\sigma$-CFDM | KID $\downarrow$ | $0.092 \pm 0.0075$ |
| | Sampling time (CPU) | 44 ms |

We report sample quality and generation time metrics, including inception scores (Salimans et al., 2016) and kernel inception distances (KID) (Bińkowski et al., 2018) between generated samples and samples from the CelebA test partition in Table 2. Our $\sigma$-CFDM results in a 15.0% improvement in KID and 32.4% improvement in inception score compared to the VAE baseline. While the VAE's sampling cost, which

amounts to the cost of generating Gaussian noise, is negligible, our method's sampling time is just 44 ms per sample. For the sake of fairness, this cost is amortized over the number of $\sigma$-CFDM samples left *after* discarding nearest neighbors to ensure novelty.

## 7 Conclusion and future work

In this work, we introduced smoothed closed-form diffusion models (smoothed CFDMs): a class of *training-free* diffusion models requiring only access to the training set at sampling time. Smoothed CFDMs leverage the availability of an exact solution to the score-matching problem—which alone does not yield generalization—and explicitly induce error by smoothing. This results in a model that generalizes by provably outputting barycenters of training points. Our method is efficient and scalable, and runs on a consumer-grade laptop with no dedicated GPU.

Our results suggest that it is possible to design SGMs that generalize without relying on neural score approximations. They also suggest that smoothness is among the inductive biases enabling neural SGMs to generalize in spite of the uninteresting global optimum of their training objective, which only allows for memorization. However, because our method generates barycenters of training points, its inductive bias is unsuitable on its own for sparsely-sampled manifolds in high-dimensional space, which one typically encounters in modern applications such as image generation. In this work, we partially address this shortfall by using our method to sample in an appropriately-structured latent space, but our sample quality lags behind that of state of the art neural SGMs. We therefore encourage further work to close this gap in sample quality, and describe some potential directions for future work below.

State of the art SGMs are typically built upon convolutional architectures with self-attention layers, which both feature unique inductive biases. Concurrent work by Kamb & Ganguli (2024) investigates the impact of locality and equivariance constraints on the optimum of the score-matching objective, and Niedoba et al. (2024) empirically investigate whether a locality bias can explain the behavior of neural denoisers. Combining these constraints with our smoothing approach and explicitly modeling the inductive biases of self-attention layers may yield further insights into the generalization of neural diffusion models and lead to new strategies for building training-free diffusion models that generalize.

Most interesting image generation tasks are *conditional*. For instance, a user may provide a text prompt and seek an image whose subject and style match the prompt. While state-of-the-art diffusion models typically employ classifier-free guidance (Ho & Salimans, 2021) to introduce conditioning information, it is unclear how to extend our training-free method to include an analogous form of guidance. On the other hand, Dhariwal & Nichol (2021)'s classifier guidance would likely be a feasible addition to our method, amounting to augmenting our velocity field (6) with the gradient of a pretrained classifier. As classifier guidance is known to improve diffusion models' sample quality, this may have the additional benefit of narrowing the gap between our samples and those generated by state-of-the-art neural methods.

## Acknowledgements

The MIT Geometric Data Processing Group acknowledges the generous support of Army Research Office grants W911NF2010168 and W911NF2110293, of National Science Foundation grant IIS2335492, from the CSAIL Future of Data program, from the MIT–IBM Watson AI Laboratory, from the Wistron Corporation, and from the Toyota–CSAIL Joint Research Center.

Christopher Scarvelis acknowledges the support of the Natural Sciences and Engineering Research Council of Canada (NSERC), funding reference number CGSD3-557558-2021, and a 2024 Exponent fellowship.

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

## A  Details on nearest-neighbor estimator of closed-form score

Karppa et al. (2022) propose an unbiased estimator of a kernel density estimate $KDE(z)$. Given a kernel function $K_h(z)$ with bandwidth $h > 0$ and a dataset $\{x_i\}_{i=1}^N$, their estimator first searches for the $K$-nearest neighbors $\{x_k\}_{k=1}^K$ of $z$ in the dataset, then draws $L$ random samples $\{x_\ell\}_{\ell=1}^L$ from the remainder of the dataset, and approximates $KDE(z)$ as follows:

$$\widehat{KDE}(z) = \frac{1}{N} \sum_{k=1}^K K_h(x_k, z) + \frac{N-K}{LN} \sum_{\ell=1}^L K_h(x_\ell, z) \tag{8}$$

This estimator is unbiased for *any* subset of points $x_k \in \{x_i\}_{i=1}^N$ drawn in the first stage. In particular, using approximate nearest-neighbors (ANNs) rather than exact nearest neighbors of $z$ increases the variance of Equation 8 but does not introduce bias.

As the closed-form score $\nabla \rho_t^*$ is the score of a Gaussian KDE $\rho_t^*$ with bandwidth $h = 2(1-t)^2$, we approximate the closed-form score using the following ratio estimator:

$$\nabla \widehat{\log \rho_t^*}(z) = \left( \frac{\widehat{\nabla \rho_t^*(z)}}{\rho_t^*(z)} \right) = \frac{\widehat{\nabla \rho_t^*}(z)}{\widehat{\rho_t^*}(z)}, \tag{9}$$

where $\widehat{\rho_t^*}(z)$ is Karppa et al. (2022)'s estimator Equation 8. Since the gradient operator is linear, both the numerator and denominator in Equation 9 are unbiased estimates of their respective terms in the closed-form score.

## B  Proofs

### B.1  Proof of Proposition 4.1

For each $k = 1, ..., N^M$, let $tC_k = (tx_{i(k,m)} : m = 1, ..., M)$ be an $M$-tuple of rescaled training points $tx_i$. (The same point $tx_i$ can appear multiple times in an $M$-tuple.)

Define the barycenters and variances of these tuples as follows:

$$t\bar{c}_k = \frac{1}{M} \sum_{m=1}^M tx_{i(k,m)}, \quad \bar{u}_k = \frac{1}{M} \sum_{m=1}^M u_{i(k,m)}, \quad \mathrm{Var}(tC_k) = \frac{1}{M} \sum_{m=1}^M \|tx_{i(k,m)} - t\bar{c}_k\|^2. \tag{10}$$

We will show that up to a constant factor, the smoothed score Equation 3 is itself the score of a mixture of $N^M$ Gaussians. Rewriting the smoothed score in gradient form, we have:

$$
\begin{aligned}
s_{\sigma,t}(z) &= \frac{1}{(1-t)^2} \frac{1}{M} \sum_{m=1}^{M} \sum_{i=1}^{N} \text{softmax} \left( -\frac{\|z - tX\|^2 + \sigma t u_{i,m}}{2(1-t)^2} \right)_i (tx_i - z) \\
&= \nabla_z \frac{1}{M} \sum_{m=1}^{M} \log \sum_{i=1}^{N} \exp \left( -\frac{\|z - tx_i\|^2 + \sigma t u_{i,m}}{2(1-t)^2} \right) \\
&= \nabla_z \frac{1}{M} \log \prod_{m=1}^{M} \sum_{i=1}^{N} \exp \left( -\frac{\|z - tx_i\|^2 + \sigma t u_{i,m}}{2(1-t)^2} \right) \\
&= \nabla_z \frac{1}{M} \log \sum_{k=1}^{N^M} \exp \left( -\frac{\sum_{m=1}^{M}(\|z - tx_{i(k,m)}\|^2 + \sigma t u_{i(k,m)})}{2(1-t)^2} \right) \\
&= \nabla_z \frac{1}{M} \log \sum_{k=1}^{N^M} \exp \left( -\frac{M \left( \|z - t\bar{c}_k\|^2 + \text{Var}(tC_k) + \sigma t \bar{u}_k \right)}{2(1-t)^2} \right) \quad (*)\\
&= \frac{1}{M} \nabla_z \log \sum_{k=1}^{N^M} \exp \left( -\frac{M(\text{Var}(tC_k) + \sigma t \bar{u}_k)}{2(1-t)^2} \right) \exp \left( -\frac{M\|z - t\bar{c}_k\|^2}{2(1-t)^2} \right) \\
&= \frac{1}{M} \nabla_z \log \sum_{k=1}^{N^M} w_k(t) \exp \left( -\frac{M\|z - t\bar{c}_k\|^2}{2(1-t)^2} \right) \\
&= \frac{1}{M} \nabla_z \log q_t(z)
\end{aligned}
$$

This shows that up to a constant factor $\frac{1}{M}$, the smoothed score $s_{\sigma,t}(z)$ is the score of a large mixture of Gaussians $q_t(z) = \sum_{k=1}^{N^M} w_k(t) \exp \left( -\frac{M \left( \|z - t\bar{c}_k\|^2 \right)}{2(1-t)^2} \right)$. The mean of each Gaussian is the barycenter $t\bar{c}_k$ of some $M$-tuple $tC_k$ of training points $tx_{i(k,m)}$, and its common covariance matrix is $\frac{(1-t)^2}{M} I$. The time-dependent mixture weights $w_k(t) \propto \exp \left( -\frac{M(\text{Var}(tC_k) + \sigma t \bar{u}_k)}{2(1-t)^2} \right)$ are decreasing in the variance of the $M$-tuples $tC_k$ but are subject to the presence of noise terms $\sigma t \bar{u}_k = \frac{\sigma t}{M} \sum_m u_{i(k,m)}$.

Finally, by expanding the gradient in $(*)$, we straightforwardly obtain:

$$
s_{\sigma,t}(z) = \frac{1}{(1-t)^2} \left( \sum_{k=1}^{N^M} \text{softmax} \left( -\frac{M}{2(1-t)^2} \left( \|z - t\bar{c}_k\|^2 + \text{Var}(tC_k) + \sigma t \bar{u}_k \right) \right)_k t\bar{c}_k - z \right)
$$

## B.2 Proof of Theorem 5.1

Define

$$
k_{\sigma,t}(z) = \sum_{k=1}^{N^M} \text{softmax} \left( -\frac{M}{2(1-t)^2} \left( \|z - t\bar{c}_k\|^2 + \text{Var}(t\tilde{C}_k) + \sigma t \bar{u}_k \right) \right)_k t\bar{c}_k \tag{11}
$$

so that $s_{\sigma,t}(z) = \frac{1}{(1-t)^2}(k_{\sigma,t}(z) - z)$. Then

$$v_{\sigma,t}(z) = \frac{1}{t}\left(z + (1-t)s_{\sigma,t}(z)\right)$$

$$= \frac{1}{t}\left(z + \frac{1}{(1-t)}(k_{\sigma,t}(z) - z)\right)$$

$$= \frac{1}{1-t}\left(\frac{1}{t}k_{\sigma,t}(z) - z\right)$$

Expanding the formula for the final Euler step using this expression for $v_{\sigma,t}(z)$ and $t_{S-1} = \frac{S-1}{S}$, we obtain:

$$z_S = z_{S-1} + \frac{1}{S}v_{\sigma,t_{S-1}}(z_{S-1})$$

$$= z_{S-1} + \frac{1}{S} \cdot \frac{1}{1 - \frac{S-1}{S}}\left(\frac{1}{\frac{S-1}{S}}k_{\sigma,\frac{S-1}{S}}(z_{S-1}) - z_{S-1}\right)$$

$$= z_{S-1} + \frac{S}{S-1}k_{\sigma,\frac{S-1}{S}}(z_{S-1}) - z_{S-1}$$

$$= \frac{S}{S-1}k_{\sigma,\frac{S-1}{S}}(z_{S-1})$$

$$= \frac{S}{S-1}\sum_{k=1}^{N^M}\text{softmax}\left(-\frac{MS^2}{2}\left(\|z_{S-1} - \frac{S-1}{S}\bar{c}_k\|^2 + \text{Var}(\frac{S-1}{S}\tilde{C}_k) + \sigma\frac{S-1}{S}\bar{u}_k\right)\right)_k \frac{S-1}{S}\bar{c}_k$$

$$\xrightarrow[S\to\infty]{} \bar{c}_{k^*}$$

In the final line, we use the fact that as $S \to \infty$, the temperature of the softmax goes to 0 and picks out a single index $k^*$ such that

$$k^* = \underset{k}{\text{argmax}} - \left(\|z_{S-1} - \bar{c}_k\|^2 + \text{Var}(C_k) + \sigma\bar{u}_k\right)$$

$$= \underset{k}{\text{argmin}}\left(\|z_{S-1} - \bar{c}_k\|^2 + \text{Var}(C_k) + \sigma\bar{u}_k\right)$$

### B.3 Proof of Theorem 5.2

We divide the proof of this theorem into three propositions. We first sketch the proof and state the propositions, and then prove each proposition in subsections below.

Our first result shows that flowing $\rho_0$ through two similar velocity fields $v_t^*, v_{\sigma,t}$ yields similar model distributions $\rho_T^*, \rho_{\sigma,T}$ at some terminal time $T$:

**Proposition B.1.** *Suppose a measure $\rho_0$ is pushed through velocity fields $v_t^*, v_{\sigma,t}$, and denote the respective pushforward measures at time $t$ by $\rho_t^*, \rho_{\sigma,t}$. Then,*

$$W_2(\rho_T^*, \rho_{\sigma,T}) \le \int_0^T \beta(t)\sqrt{\underset{z\sim\rho_t^*}{\mathbb{E}}\|v_t^*(z) - v_{\sigma,t}(z)\|^2}\,dt \tag{12}$$

*where $\beta(t) := \exp\left(\int_t^T L_{v_s^*}\,ds\right)$ and $L_{v_s^*} \ge 0$ is the Lipschitz constant of $v_s^*$.*

The result above applies to any two velocity fields, subject to some weak regularity conditions. To apply this result to the unsmoothed and smoothed velocity fields $v_t^*$ and $v_{\sigma,t}$, we bound $\beta(t)$ and $\mathbb{E}\|v_t^*(z) - v_{\sigma,t}(z)\|^2$ in terms of $\sigma$:

**Proposition B.2.** *Let $v_t^*$ be velocity field of an unsmoothed CFDM, and let $v_{\sigma,t}^*$ be the velocity field Equation 6 of the corresponding $\sigma$-CFDM. Then,*

$$\beta(t) \leq \exp\left(\frac{D^2(2T-1)}{(1-T)^2}\right) \cdot \left(\frac{1-t}{1-T}\right)^2 \tag{13}$$

*and*

$$\sqrt{\mathop{\mathbb{E}}_{z \sim \rho_t^*} \|v_t^*(z) - v_t(z)\|^2} \leq C_1 \frac{\sigma t}{2(1-t)^3} \tag{14}$$

*where $D$ is the diameter of the training data and $C_1$ is a constant depending on the training data and the distribution $p_u$ of the scalar noise $u_{i,m}$ perturbing the distance weights in Equation 4.*

Combining these results, we obtain the following bound on $W_2(\rho_T^*, \rho_{\sigma,T})$:

$$W_2(\rho_T^*, \rho_{\sigma,T}) = O(\sigma). \tag{15}$$

This shows that one can approximate a $\sigma$-CFDM's model samples at some time $T > 0$ by model samples from its corresponding unsmoothed CFDM (i.e. a mixture of Gaussians) when the smoothing parameter $\sigma$ is small.

We now show that flowing two similar distributions $\rho_T^*$ and $\rho_{\sigma,T}$ through a $\sigma$-CFDM's velocity field from time $T$ to $1 - \epsilon$ yields similar terminal distributions $\rho_{\sigma,1-\epsilon}^T, \rho_{\sigma,1-\epsilon}^0$. Following De Bortoli (2022), we stop sampling at time $1 - \epsilon$ for some truncation parameter $\epsilon > 0$ to account for the fact that the smoothed score $s_{\sigma,t}$ blows up as $t \to 1$ due to division by $(1-t)^2$.

**Proposition B.3.** *Suppose $\rho_T^*$ and $\rho_{\sigma,T}$ are pushed through the velocity field $v_{\sigma,t}$ of a $\sigma$-CFDM, and let $\rho_{\sigma,1-\epsilon}^T, \rho_{\sigma,1-\epsilon}^0$ denote their respective terminal distributions at time $1 - \epsilon$. Then*

$$W_2(\rho_{\sigma,1-\epsilon}^T, \rho_{\sigma,1-\epsilon}^0) \leq O\left(\left(\frac{1-T}{\epsilon}\right)^2 \exp\left(\frac{D^2(1-2\epsilon)}{\epsilon^2}\right)\right) W_2(\rho_T^*, \rho_{\sigma,T}), \tag{16}$$

*where $D$ is the diameter of the training data.*

By combining Equation 15 and Equation 16 and treating $T$ and the truncation parameter $\epsilon$ as fixed, we finally obtain a global upper bound on $W_2(\rho_{\sigma,1-\epsilon}^T, \rho_{\sigma,1-\epsilon}^0)$:

$$W_2(\rho_{\sigma,1-\epsilon}^T, \rho_{\sigma,1-\epsilon}^0) = O(\sigma) \tag{17}$$

where $\rho_{\sigma,1-\epsilon}^T$ is the model distribution obtained by starting sampling at $T > 0$ with samples from the unsmoothed CFDM and $\rho_{\sigma,1-\epsilon}^0$ is true model distribution of the $\sigma$-CFDM.

### B.3.1 Proof of Proposition B.1

Our proof for this proposition employs techniques similar to those used to prove Theorem 1 and Proposition 1 in Kwon et al. (2022).

We begin with the following well-known result (Santambrogio, 2015, Corollary 5.25):

Suppose that two measures $\rho^*$ and $\rho$ are each pushed through velocity fields $v_t^*, v_t$ respectively and denote the pushforward measures at time $t$ by $\rho_t^*, = \rho_t^*$. Then:

$$\frac{1}{2}\frac{\mathrm{d}}{\mathrm{d}t} W_2^2(\rho_t^*, \rho_t) = \mathop{\mathbb{E}}_{(x,y) \sim \gamma_t} \langle y - x, v_t^*(y) - v_t(x) \rangle \tag{18}$$

where $\gamma_t$ is the $W_2$ coupling between $\rho_t^*$ and $\rho_t$.

For any $x, y$ we can use Cauchy-Schwarz and the triangle inequality to obtain the following bound:

$$\langle y - x, v_t^*(y) - v_t(x) \rangle \leq \|y - x\| \cdot (\|v_t^*(y) - v_t^*(x)\| + \|v_t^*(x) - v_t(x)\|) \tag{19}$$

We can then bound $\|v_t^*(y) - v_t^*(x)\|$ in terms of maximum of the Jacobian $Dv_t^*$ of $v_t^*$ on the line segment $[x, y] := \{ty + (1 - t)x : 0 \leq t \leq 1\}$ to obtain:

$$\langle y - x, v_t^*(y) - v_t(x) \rangle \leq \left( \max_{p \in [x,y]} \|Dv_t^*\| \right) \|y - x\|^2 + \|y - x\| \cdot \|v_t^*(x) - v_t(x)\| \tag{20}$$

This constant is in turn upper-bounded by the Lipschitz constant $L_{v_t^*}$ of $v_t^*$ on the convex hull of $\operatorname{supp}(\rho_t^*) \cup \operatorname{supp}(\rho_t)$, so we in fact have:

$$\langle y - x, v_t^*(y) - v_t(x) \rangle \leq L_{v_t^*} \|y - x\|^2 + \|y - x\| \cdot \|v_t^*(x) - v_t(x)\| \tag{21}$$

Adding $\underset{(x,y) \sim \gamma_t}{\mathbb{E}}$ back in, we get:

$$\begin{aligned}
\frac{1}{2} \frac{\mathrm{d}}{\mathrm{d}t} W_2^2(\rho_t^*, \rho_t) &= \underset{(x,y) \sim \gamma_t}{\mathbb{E}} \langle y - x, v_t^*(y) - v_t(x) \rangle \\
&\leq L_{v_t^*} \underset{(x,y) \sim \gamma_t}{\mathbb{E}} \|y - x\|^2 + \underset{(x,y) \sim \gamma_t}{\mathbb{E}} \|y - x\| \cdot \|v_t^*(x) - v_t(x)\| \\
&= L_{v_t^*} W_2^2(\rho_t^*, \rho_t) + \underset{(x,y) \sim \gamma_t}{\mathbb{E}} \|y - x\| \cdot \|v_t^*(x) - v_t(x)\| \\
&\leq L_{v_t^*} W_2^2(\rho_t^*, \rho_t) + \sqrt{\underset{(x,y) \sim \gamma_t}{\mathbb{E}} \|y - x\|^2} \cdot \sqrt{\underset{(x,y) \sim \gamma_t}{\mathbb{E}} \|v_t^*(x) - v_t(x)\|^2} \\
&= L_{v_t^*} W_2^2(\rho_t^*, \rho_t) + W_2(\rho_t^*, \rho_t) \cdot \sqrt{\underset{x \sim \rho_t^*}{\mathbb{E}} \|v_t^*(x) - v_t(x)\|^2}
\end{aligned}$$

where we use Cauchy-Schwarz for random variables in passing from the third to fourth lines and then the fact that $\rho_t^*$ is one of the marginals of $\gamma_t$. Using the chain rule on the LHS and cancelling a factor of $W_2(\rho_t^*, \rho_t)$ from both sides, we obtain the following differential inequality:

$$\frac{\mathrm{d}}{\mathrm{d}t} W_2(\rho_t^*, \rho_t) \leq L_{v_t^*} W_2(\rho_t^*, \rho_t) + \sqrt{\underset{x \sim \rho_t^*}{\mathbb{E}} \|v_t^*(x) - v_t(x)\|^2} \tag{22}$$

We can now solve the differential inequality Equation 22 to obtain:

$$W_2(\rho_T^*, \rho_T) \leq \int_0^T \beta(t) \sqrt{\underset{x \sim \rho_t^*}{\mathbb{E}} \|v_t^*(x) - v_t(x)\|^2} \mathrm{d}t \tag{23}$$

where

$$\beta(t) := \exp\left( \int_t^T L_{v_s^*} \mathrm{d}s \right) \tag{24}$$

### B.3.2 Proof of Proposition B.2

We first estimate $\beta(t) = \exp\left(\int_t^T L_{v_s^*} ds\right)$. As

$$v_t^*(z) = \frac{1}{t(1-t)} k_t^*(z) - \frac{1}{1-t} z, \tag{25}$$

we can bound its Lipschitz constant by $L_{v_t^*} \leq 2 \max\left\{ \frac{1}{t(1-t)} L_{k_t^*}, \frac{1}{1-t} \right\}$. Our next step is therefore to bound $L_{k_t^*}$. We will do so by bounding the spectral norm of the Jacobian $\|Jk_t^*(z)\|_2$ of $k_t^*$ for any $z \in \mathbb{R}^D$.

To this end, we first note that for any $z \in \mathbb{R}^D$, this Jacobian has the form of a weighted covariance matrix:

$$Jk_t^*(z) = \frac{1}{(1-t)^2} \sum_{i=1}^N w_i(z)(tx_i - k_t^*(z))(tx_i - k_t^*(z))^\top,$$

where $w_i(z)$ is the $i$-th component of the weight vector $\text{softmax}\left(-\frac{\|z - tX\|^2}{2(1-t)^2}\right)$. To upper bound its spectral norm, we first apply the triangle inequality and use the absolute homogeneity of norms:

$$\begin{aligned}
\|Jk_t^*(z)\|_2 &= \left\| \frac{1}{(1-t)^2} \sum_{i=1}^N w_i(z)(tx_i - k_t^*(z))(tx_i - k_t^*(z))^\top \right\|_2 \\
&\leq \frac{1}{(1-t)^2} \sum_{i=1}^N w_i(z) \|(tx_i - k_t^*(z))(tx_i - k_t^*(z))^\top\|_2.
\end{aligned}$$

But it is well-known that the spectral norm of a rank-1 matrix $uv^\top$ is given by $\|uv^\top\|_2 = \|u\|_2 \|v\|_2$, so in fact

$$\|Jk_t^*(z)\|_2 \leq \frac{1}{(1-t)^2} \sum_{i=1}^N w_i(z) \|tx_i - k_t^*(z)\|_2^2. \tag{26}$$

Now, because $k_t^*(z)$ is a convex combination of the $tx_i$, it lies in the convex hull of the $tx_i$. The diameter of a convex hull is bounded by the diameter of its extreme points, so if $D = \text{diam}(\{x_i\})$ and $tD = \text{diam}(\{tx_i\})$, then $\|tx_i - k_t^*(z)\|_2^2 \leq (tD)^2$. Substituting this back into equation 26 and using the fact that the $w_i(z)$ sum to 1 for any $z$, we obtain the bound $\|Jk_t^*(z)\|_2 \leq \left(\frac{tD}{1-t}\right)^2$. This bound holds for all $z \in \mathbb{R}^D$, so it follows that $L_{k_t^*} \leq \left(\frac{tD}{1-t}\right)^2$.

Hence $L_{v_t^*} \leq 2 \max\left\{ \frac{tD^2}{(1-t)^3}, \frac{1}{1-t} \right\}$.

We now use this bound on $L_{v_t^*}$ to estimate $\beta(t)$. Let $\bar{s} \in [0, T]$ denote the time from which $\frac{\bar{s}D^2}{(1-\bar{s})^3} \geq \frac{1}{1-\bar{s}}$. Then $\bar{s} = \frac{1}{2}(D^2 - D\sqrt{D^2 + 4} + 2)$. Decomposing the integral that defines $\log \beta(t)$, we obtain:

$$\int_t^T L_{v_s^*} \mathrm{d}s = \int_t^{\bar{s}} L_{v_s^*} \mathrm{d}s + \int_{\bar{s}}^T L_{v_s^*} \mathrm{d}s$$

$$\leq 2 \int_t^{\bar{s}} \frac{1}{1-s} \mathrm{d}s + 2D^2 \int_{\bar{s}}^T \frac{s}{(1-s)^3} \mathrm{d}s$$

$$= 2 \log \left( \frac{1-t}{1-\bar{s}} \right) + D^2 \left( \frac{2T-1}{(1-T)^2} - \frac{2\bar{s}-1}{(1-\bar{s})^2} \right)$$

$$\leq 2 \log \left( \frac{1-t}{1-T} \right) + D^2 \left( \frac{2T-1}{(1-T)^2} - 4\frac{D^2 - D\sqrt{D^2+4}+1}{(D^2 - D\sqrt{D^2+4})^2} \right)$$

Substituting this bound into $\beta(t) = \exp(\int_t^T L_{v_s^*})$ and simplifying, we obtain:

$$\beta(t) \leq C(T) \cdot \left( \frac{1-t}{1-T} \right)^2 \tag{27}$$

where $C(T) = \exp \left( D^2 \left( \frac{2T-1}{(1-T)^2} - 4\frac{D^2 - D\sqrt{D^2+4}+1}{(D^2 - D\sqrt{D^2+4})^2} \right) \right) = \exp \left( \frac{D^2(2T-1)}{(1-T)^2} \right)$ and $D^2 =: C_0$ depends only on the training data.

We now estimate $\sqrt{\mathbb{E}_{x \sim \rho_t^*} \|v_t^*(x) - v_t(x)\|^2}$.

We first observe that $v_t^*(z) - v_t(z) = \frac{1}{t(1-t)}(k_t^*(z) - k_t(z))$. Once again letting $X \in \mathbb{R}^{D \times N}$ be the matrix of training data and $w^*(z) = \mathrm{softmax} \left( -\frac{\|z-tx\|^2}{2(1-t)^2} \right) \in \mathbb{R}^N$, $\tilde{w}_m(z) = \mathrm{softmax} \left( -\frac{\|z-tx\|^2 + \sigma t u_{i,m}}{2(1-t)^2} \right) \in \mathbb{R}^N$ be the vector of weights, we have that $k_t^*(z) = tXw^*(z)$ and hence

$$\|v_t^*(z) - v_t(z)\| = \frac{1}{1-t}\|X(w^*(z) - \frac{1}{M}\sum_{m=1}^M \tilde{w}_m(z))\| \leq \frac{1}{1-t}\|X\| \cdot \frac{1}{M}\sum_{m=1}^M \|w^*(z) - \tilde{w}_m(z)\|. \tag{28}$$

Once again using the Lipschitz continuity of $w(z)$, we obtain the bound

$$\|w^*(z) - \tilde{w}_m(z)\| \leq \frac{\sigma t u_{i,m}}{2(1-t)^2}, \tag{29}$$

and by substituting this into our bound for $\|v_t^*(z) - v_t(z)\|$, we obtain:

$$\|v_t^*(z) - v_t(z)\|^2 \leq \frac{t^2\sigma^2\|X\|^2\bar{u}_i^2}{4(1-t)^6}, \tag{30}$$

where $\bar{u}_i = \frac{1}{M}\sum_m u_{i,m}$. As this bound holds for all $z$, it also holds in expectation, so we finally conclude that

$$\sqrt{\mathbb{E}_{x \sim \rho_t^*} \|v_t^*(x) - v_t(x)\|^2} \leq \frac{t\sigma\|X\|\bar{u}_i}{2(1-t)^3}. \tag{31}$$

### B.3.3 Proof of Proposition B.3

We now begin with the differential inequality

$$\frac{\mathrm{d}}{\mathrm{d}t}W_2(\rho_t^*, \rho_t) \leq L_{v_t^*}W_2(\rho_t^*, \rho_t) + \sqrt{\mathbb{E}_{x \sim \rho_t^*} \|v_t^*(x) - v_t(x)\|^2}, \tag{32}$$

that we derived in the proof of Proposition B.1, which bounds the rate of change in $W_2(\rho_t^*, \rho_t)$ when flowing $\rho_t^*$ and $\rho_t$ through two velocity fields $v_t^*$ and $v_t$, respectively. As we now consider the case where $\rho_T^*$ and $\rho_{\sigma,t}$ both flow through the smoothed velocity field $v_{\sigma,t}$ from time $T$ to $1 - \epsilon$, $\sqrt{\underset{x \sim \rho_t^*}{\mathbb{E}} \|v_t^*(x) - v_t(x)\|^2} = 0$ and the differential inequality becomes:

$$\frac{\mathrm{d}}{\mathrm{d}t} W_2(\rho_t^*, \rho_{\sigma,t}) \leq L_{v_{\sigma,t}} W_2(\rho_t^*, \rho_{\sigma,t}). \tag{33}$$

Solving this differential inequality, we obtain

$$W_2(\rho_{\sigma,1-\epsilon}^T, \rho_{\sigma,1-\epsilon}^0) := W_2(\rho_{1-\epsilon}^*, \rho_{\sigma,1-\epsilon}) \leq \tilde{\beta}(T) W_2(\rho_T^*, \rho_{\sigma,T}) \tag{34}$$

where $\tilde{\beta}(T) = \exp\left(\int_T^{1-\epsilon} L_{v_{s,\sigma}} \mathrm{d}s\right)$. Using the same bounds as in our proof of Proposition B.2 while noting that $v_{\sigma,t}$ is at least as smooth as $v_t^*$, we obtain

$$
\begin{aligned}
\tilde{\beta}(T) &\leq \exp\left(2\log(\frac{1-T}{\epsilon}) + D^2\left(\frac{1-2\epsilon}{\epsilon^2} - 4\frac{D^2 - D\sqrt{D^2+4}+1}{(D^2 - D\sqrt{D^2+4})^2}\right)\right) \\
&= \left(\frac{1-T}{\epsilon}\right)^2 \exp\left(D^2\left(\frac{1-2\epsilon}{\epsilon^2} - 4\frac{D^2 - D\sqrt{D^2+4}+1}{(D^2 - D\sqrt{D^2+4})^2}\right)\right) \\
&= O\left(\left(\frac{1-T}{\epsilon}\right)^2 \exp\left(\frac{D^2(1-2\epsilon)}{\epsilon^2}\right)\right)
\end{aligned}
$$

where $D^2$ is the diameter of the training data, which we treat as constant for a given CFDM. Substituting this into Equation 34, we obtain

$$W_2(\rho_{\sigma,1-\epsilon}^T, \rho_{\sigma,1-\epsilon}^0) \leq O\left(\left(\frac{1-T}{\epsilon}\right)^2 \exp\left(\frac{D^2(1-2\epsilon)}{\epsilon^2}\right)\right) W_2(\rho_T^*, \rho_{\sigma,T}). \tag{35}$$

### B.4 Proof of Proposition 5.3

We showed in Theorem 5.1 (see Appendix B.2) that as the number of sampling steps $S \to \infty$, the samples from a smoothed CFDM converge towards barycenters $z_S = \bar{c}_{k^*}$ of $M$-tuples of training points for indices $k^*$ such that:

$$k^*(z_{S-1}) = \underset{k}{\mathrm{argmax}} - \left(\|z_{S-1} - \bar{c}_k\|^2 + \mathrm{Var}(\tilde{C}_k) + \sigma \bar{u}_k\right) \tag{36}$$

Using an equivalent expression for $\tilde{k}_{\sigma,t}$, these barycenters can also be written as

$$z_S = \bar{c}_{k^*} = \frac{1}{M} \sum_{m=1}^M x_{i^*(z_{S-1},m)}, \tag{37}$$

where

$$i^*(z_{S-1}, m) = \operatorname*{argmax}_i - \left( \|z_{S-1} - x_i\|^2 + \sigma u_{i,m} \right)$$

$$= \operatorname*{argmax}_i - \left( \frac{1}{\sigma} \|z_{S-1} - x_i\|^2 + u_{i,m} \right)$$

If $u_{i,m} \sim \text{Gumbel}(0,1)$, then by applying the Gumbel max-trick, we conclude that $i^*(z_{S-1}, m) \sim$ Categorical($\pi_\sigma^i$). This is a distribution over the indices $i = 1, ..., N$ of training samples, with event probabilities given by

$$\pi_\sigma^i = \text{softmax} \left( -\frac{1}{\sigma} \|z_{S-1} - x_i\|^2 \right)_i \tag{38}$$

If we represent $x_{i^*}$ as $Xe_{i^*}$, where $X \in \mathbb{R}^{D \times N}$ is the matrix whose $i$-th column is training sample $x_i$ and $e_{i^*} \in \mathbb{R}^N$ is the $i^*$-th standard basis vector, then

$$z_S = \frac{1}{M} \sum_{m=1}^M x_{i^*(z_{S-1}, m)}$$

$$= \frac{1}{M} \sum_{m=1}^M (Xe_{i^*})$$

$$= \frac{1}{M} X \sum_{m=1}^M e_{i^*}$$

$$= \frac{1}{M} X I_\sigma$$

But $I_\sigma := \sum_{m=1}^M e_{i^*}$ is a realization of a Multinomial($\pi_\sigma, M$) random variable; this fact completes the proof of Proposition 5.3.

## C  Additional Experimental Details and Results

In this appendix, we provide details for our pixel space and latent space image generation experiments.

### C.1  Pixel space DDPM training details

Our training data is drawn from the dataset `huggan/smithsonian_butterflies_subset`, which is publicly available on `huggingface` and contains 1000 images of butterflies. We extract RGB images from the `image` column of their dataset and reshape them to $128 \times 128$ before using them in our experiments. We construct an 80/20 train-test split and use the train partition to train the DDPM and to construct our $\sigma$-CFDM, and use the test partition to compute metrics.

We use the DDPM implemented in the `lucidrains` library `denoisingdiffusionpytorch` as our baseline. We use their UNet with `dim_mults=(1, 2, 4, 8)` as a backbone. We use 1000 time steps during training, and use DDIM sampling with 100 time steps during sample generation. We train the diffusion model with a batch size of 8 at a learning rate of $5 \times 10^{-5}$ for 20,000 iterations.

We center and normalize the training data to lie in the unit ball before using it to construct our $\sigma$-CFDM. We set $M = 2$ and $\sigma = 0.1$ for this experiment, and compute the smoothed score exactly rather than using our nearest neighbor-based estimator from Section 5.4 due to this dataset's relatively small size. We start sampling at $T = 0.98$ and use step size 0.01. We filter out model samples whose Euclidean distance is within

$10^{-6}$ of their nearest neighbor in the training set; with these hyperparameters, roughly 60% of the model samples remain after this filtering step.

We compute our metrics using the `torchmetrics` implementation of the kernel inception distance (KID) and inception score. We compute KID scores with `subset_size=50` between 500 randomly-chosen images from the test partition and our CFDM and DDPM samples.

### C.2  CelebA latent space generation details

Our method uses the nuclear norm-regularized autoencoder from Scarvelis & Solomon (2024). This autoencoder operates on $256 \times 256$ images from the CelebA dataset. To reduce the memory and compute costs of our autoencoder, we perform a discrete cosine transform (DCT) using the `torch-dct` package and keep only the first 80 DCT coefficients. We then pass these coefficients into the autoencoder.

The autoencoder consists of an encoder $f_\theta$ followed by a decoder $g_\phi$. The encoder $f_\theta$ is parametrized as a two-layer MLP with 10,000 hidden units; the latent space is 700-dimensional. The decoder $g_\phi$ consists of a two-layer MLP with 10,000 hidden units and $3 * 80 * 80 = 19200$ output dimensions, followed by an inverse DCT, and finally a UNet. We set the regularization parameter to $\eta = 4$ (see (Scarvelis & Solomon, 2024, Appendix B.3) for details on the training objective) and use a log-cosh reconstruction loss (Chen et al., 2019) for improved sample quality. We train for 100 epochs at a learning rate of $10^{-4}$ using the AdamW optimizer (Loshchilov & Hutter, 2017).

We then sample our $\sigma$-CFDM in the latent space of this pre-trained autoencoder. We center and normalize the training data to lie in the unit ball before using it to construct our $\sigma$-CFDM. We set $M = 2$ and $\sigma = 0.025$ for this experiment and use the nearest neighbor-based score estimator described in Section 5.4. We start sampling at $T = 0.99$ and use step size 0.01. We filter out model samples whose Euclidean distance is within $10^{-6}$ of their nearest neighbor in the training set; with these hyperparameters, roughly 34% of the model samples remain after this filtering step.

Our baseline is a VAE with the same architecture as the nuclear norm-regularized autoencoder and the same log-cosh reconstruction loss. We set the regularization strength at $10^{-3}$ and train for 100 epochs at a learning rate of $10^{-4}$ using the AdamW optimizer. At sampling time, we decode Gaussian samples drawn from $\mathcal{N}(0, 10I)$; we find that this results in improved sample quality relative to sampling from a standard normal distribution.

We compute our metrics using the `torchmetrics` implementation of the kernel inception distance (KID) and inception score. We compute KID scores with `subset_size=50` between 500 randomly-chosen images from the test partition and our CFDM and DDPM samples.

## D   Impact of $M$ on model samples

In this appendix, we demonstrate the impact of $M$ – the number of noise samples used to computed the smoothed score Equation 3 – on a $\sigma$-CFDM's model samples. In Figure 8, we use a simple training set of 2 points (in blue), fix $\sigma = 1$, generate 100 $\sigma$-CFDM samples (in red) for different values of $M$. Note in particular that for large values of $M$, the model samples cluster around the centroid of the two training points. We conjecture that this phenomenon may be explained by the law of large numbers: As $M \to \infty$, $\frac{1}{M} \sum_{m=1}^{M} k_t(x + \sigma \epsilon_m) \to \mathbb{E}_\epsilon k_t(x + \sigma \epsilon)$, which is a deterministic quantity lying on the line segment connecting the two training points. In this regime, the reasoning used in the proof of Theorem 5.1 suggests that conditional on the second-to-last sampling iterate $z_{S-1}$, the output of a $\sigma$-CFDM becomes deterministic and all randomness in the model samples originates from $z_{S-1}$.

In Figure 9, we carry out a similar experiment with a training set consisting of 500 samples from the checkerboard distribution and $\sigma = 0.3$. Note that for large values of $M$, the model samples recede from boundary of the convex hull of the training data; we conjecture that this is an instance of the same phenomenon as in Figure 8.

We finally experiment with the impact of $M$ on the latent sampling algorithm introduced in Section 6.4. We fix all hyperparameters but $M$ to the values described in Appendix C.2 and use our $\sigma$-CFDM to sample with $M \in \{2, 4, 6, 8\}$. We depict grids of decoded samples in Figure 10 and report sample quality metrics in Table 3. Using small values of $M$ tends to yield increased sample diversity while keeping sampling costs low, whereas large values of $M$ yield a greater proportion of samples which qualitatively resemble barycenters of face images. This is also likely an instance of the phenomenon illustrated in Figure 8, in which model samples cluster around centroids of training samples for large values of $M$.

Table 3: Metrics for $\sigma$-CFDM sample quality and generation time in latent space as a function of $M$.

| Value of $M$ | Metric | CelebA |
|---|---|---|
| | Inception score ↑ | $2.18 \pm 0.22$ |
| $M = 2$ | KID ↓ | $0.091 \pm 0.0071$ |
| | Sampling time (CPU) | 41 ms |
| | Inception score ↑ | $2.13 \pm 0.16$ |
| $M = 4$ | KID ↓ | $0.095 \pm 0.0077$ |
| | Sampling time (CPU) | 52 ms |
| | Inception score ↑ | $2.03 \pm 0.17$ |
| $M = 6$ | KID ↓ | $0.098 \pm 0.0088$ |
| | Sampling time (CPU) | 97 ms |
| | Inception score ↑ | $2.11 \pm 0.19$ |
| $M = 8$ | KID ↓ | $0.095 \pm 0.0073$ |
| | Sampling time (CPU) | 368 ms |

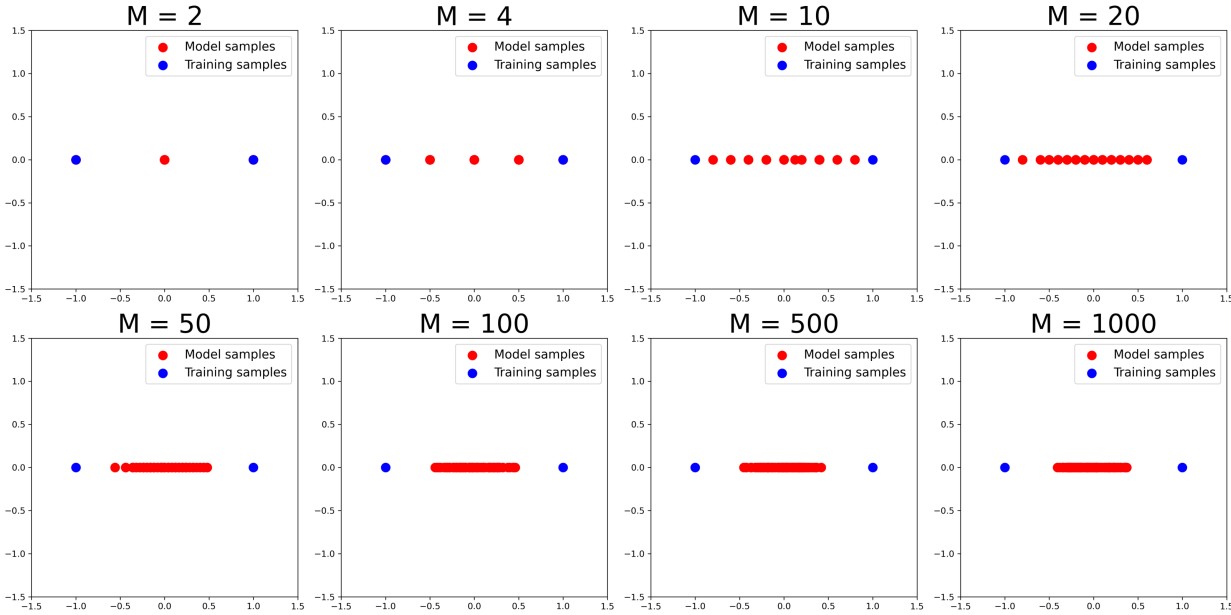

Figure 8: $\sigma$-CFDM samples (in red) generated given two training points (in blue) for various $M$.

## E   Comparison of Gaussian and Gumbel noise for latent space sampling

In this appendix, we briefly illustrate our method's robustness to the distribution of noise used for smoothing the closed-form score. We use our $\sigma$-CFDM to sample CelebA images in latent space using the same hyperparameters as described in Section C.2, and compare the effects of smoothing the closed-form score with Gaussian noise and with Gumbel noise whose first two moments match those of the Gaussian noise. We

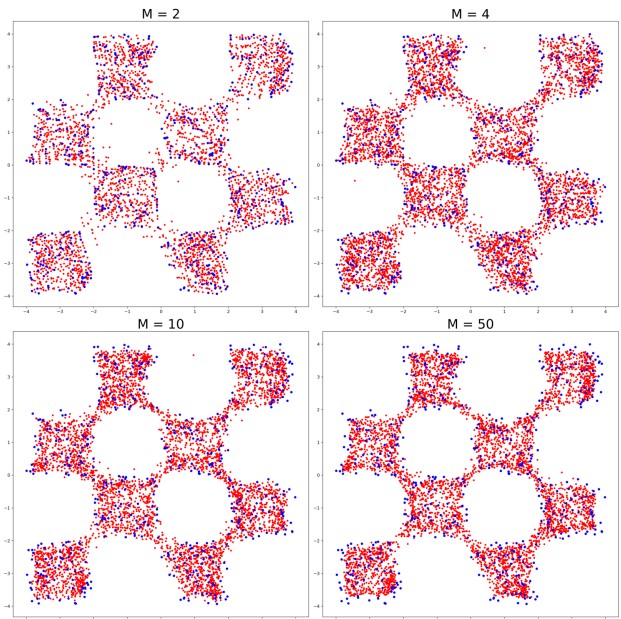

Figure 9: $\sigma$-CFDM samples (in red) generated given 500 training samples from the checkerboard distribution (in blue) for various $M$.

depict grids of decoded samples in Figure 11 and report sample quality metrics in Table 4. Our method is robust to the distribution of smoothing noise, with Gaussian and Gumbel noise resulting in similar sample quality metrics and qualitatively similar samples.

Table 4: Metrics for $\sigma$-CFDM sample quality and generation time in latent space when smoothing with Gaussian and Gumbel noise with matched mean and covariance.

| Noise distribution | Metric | CelebA |
|:---:|:---:|:---:|
| | Inception score ↑ | $2.18 \pm 0.22$ |
| Gaussian | KID ↓ | $0.091 \pm 0.0071$ |
| | Sampling time (CPU) | 41 ms |
| | Inception score ↑ | $2.22 \pm 0.19$ |
| Gumbel | KID ↓ | $0.092 \pm 0.0082$ |
| | Sampling time (CPU) | 50 ms |

# F  Adding noise to the velocity field does not induce generalization

The proof of Theorem 5.1 shows that in the limit of small step sizes, a $\sigma$-CFDM outputs barycenters of training samples in its final sampling iteration. This is true regardless of the position of the second-to-last iterate $z_{S-1}$. Consequently, adding noise to the velocity field at each sampling step, as is typical for the Euler-Maruyama scheme to simulate stochastic differential equations (SDEs), does not fundamentally change the result of Theorem 5.1: "SDE sampling" for a $\sigma$-CFDM would still output barycenters of training samples, provided the noise variance vanishes as $t \to 1$. (If the noise variance does not vanish at the final sampling iteration, then the sampler will return noisy samples as some noise remains present in the final iteration.)

Furthermore, adding noise to an unsmoothed CFDM's velocity field is insufficient to induce generalization. Theorem 5.1 also shows that the output $z_S$ of our sampler is of the form $\frac{S}{S-1} k_{\sigma, \frac{S-1}{S}}(z_{S-1})$. This would not change if the velocity field were augmented with noise that vanishes in the final sampling iteration; only $z_{S-1}$ would change. When the smoothing parameter $\sigma = 0$, $k_{\sigma, \frac{S-1}{S}}(z_{S-1}) = k_{\frac{S-1}{S}}(z_{S-1}) =$

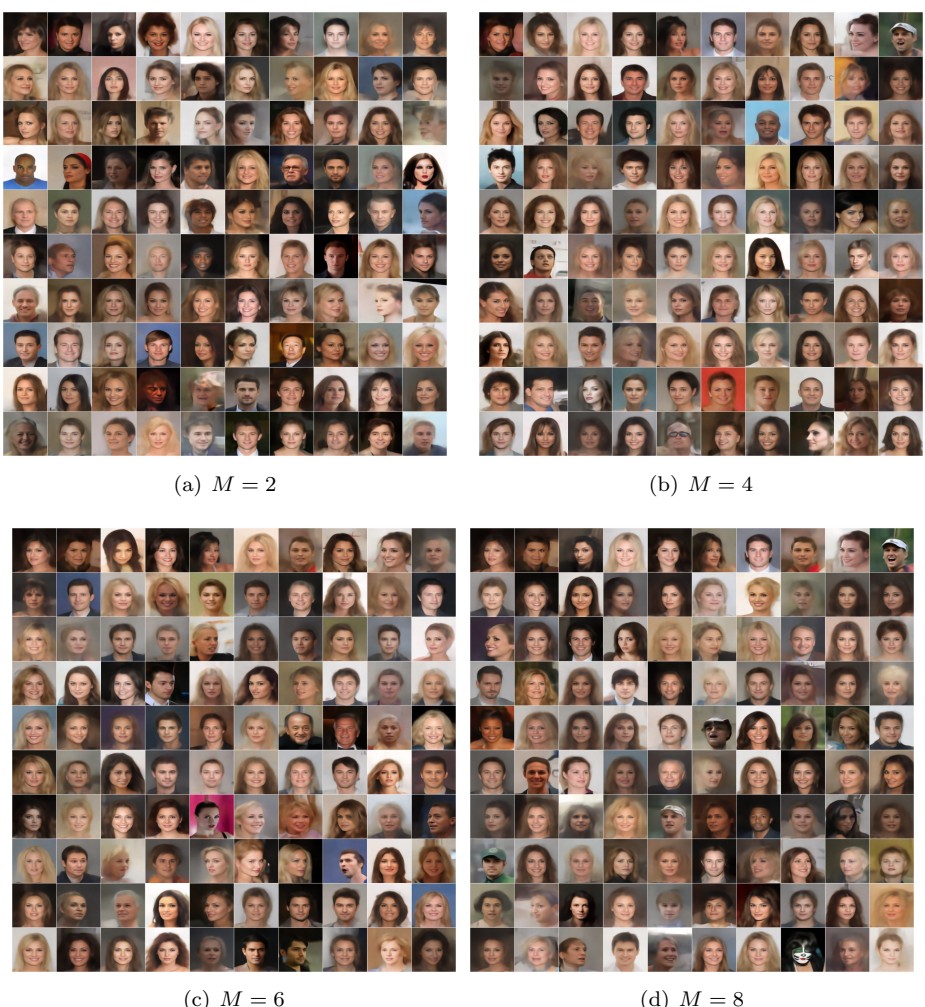

(a) $M = 2$

(b) $M = 4$

(c) $M = 6$

(d) $M = 8$

Figure 10: $\sigma$-CFDM samples drawn in latent space with $M \in \{2, 4, 6, 8\}$.

$\sum_{i=1}^{N} \text{softmax} \left( -\frac{\|z - \frac{S-1}{S} X\|^2}{2(1 - \frac{S-1}{S})^2} \right)_i \frac{S-1}{S} x_i$. For sufficiently small step sizes, the temperature of this softmax is nearly 0, which implies that the unsmoothed CFDM outputs a training sample in its final iteration, regardless of the penultimate sample $z_{S-1}$. It follows that augmenting the velocity field $v_t$ of an unsmoothed CFDM with noise does not induce generalization.

We demonstrate this empirically in Figure 12, in which we sample from an unsmoothed CFDM constructed from 500 empirical samples from the 2D checkerboard distribution, and from the same CFDM whose velocity field at time $t$ is augmented with Gaussian noise with covariance $\sqrt{0.1}(1 - t)I$. The step size in these experiments is $10^{-2}$. Adding noise to the velocity field does not induce generalization, and the model samples are numerically identical to training samples in each case: the chamfer distance between the noiseless CFDM's model samples and the training samples is $6.41 \times 10^{-4}$, and the chamfer distance between the noisy CFDM's model samples and the training samples is $6.59 \times 10^{-4}$.

## G   Impact of step size on generalization

In this section, we demonstrate empirically that an unsmoothed CFDM memorizes its training data, even when sampled with relatively few iterations.

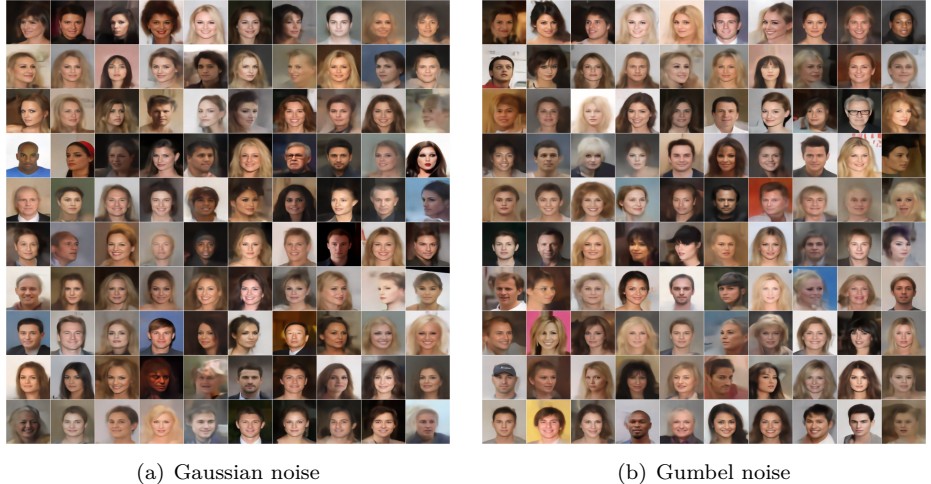

(a) Gaussian noise

(b) Gumbel noise

Figure 11: $\sigma$-CFDM samples drawn in latent space using Gaussian noise (left) and Gumbel noise (right) with matched means and covariances to smooth the closed-form score.

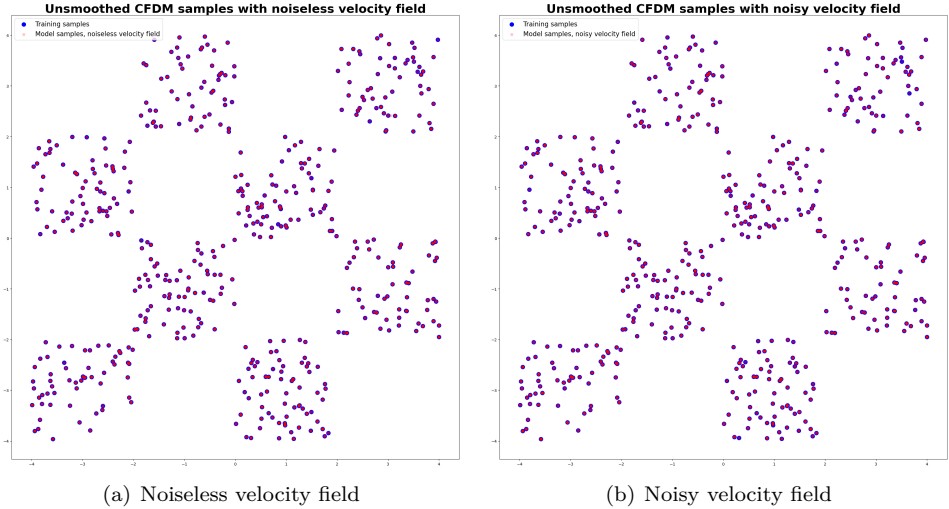

(a) Noiseless velocity field

(b) Noisy velocity field

Figure 12: Augmenting an unsmoothed CFDM's velocity field with Gaussian noise that vanishes in the final step does not induce generalization.

We first show that sampling an unsmoothed CFDM with large step sizes may in principle lead to generalization. To do so, we employ similar arguments as in the proof of Theorem 5.1 in Appendix B.2. In particular, if $\sigma = 0$, then the smoothed velocity field $v_{\sigma,t}(z)$ can be expressed as $v_{0,t}(z) = v_t(z) = \frac{1}{1-t}\left(\frac{1}{t}k_t(z) - z\right)$. By expanding the formula for the final Euler step using this expression for $v_t(z)$ and $t_{S-1} = \frac{S-1}{S}$ in the same way as in Appendix B.2 but substituting the softmax weights from the definition of $k_t(z)$ in equation 2, one sees that the final iterate $z_S$ will be of the form

$$z_S = \sum_{i=1}^{N} \text{softmax}\left(-\frac{S^2 \|z - (\frac{S}{S-1})X\|^2}{2}\right)_i x_i.$$

If the number of sampling iterations $S$ is small, then the temperature of this softmax may be sufficiently large that $z_S$ is a non-trivial convex combination of training samples. However, we demonstrate empirically

that this does not occur in practice, even for relatively small $S$. We construct an unsmoothed CFDM using 500 samples from a 100-dimensional standard Gaussian distribution to mimic the high-dimensional data that is typical of real-world applications, sample it with a number of iterations ranging from $S = 1$ to $S = 100$, and compute the chamfer distance between the model samples and the training samples for each value of $S$. We depict the result of this experiment in Figure 13. An unsmoothed CFDM memorizes its training data when sampled with as few as 5 iterations, demonstrating that in practice, large step sizes alone are insufficient to cause a CFDM to generalize.

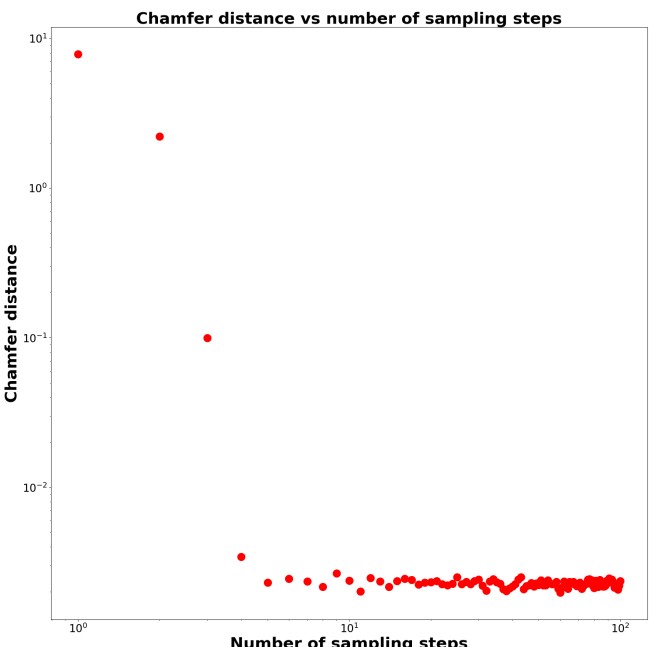

Figure 13: An unsmoothed CFDM memorizes its training data when sampled with as few as 5 iterations.

## H    Comparing neural SGMs to $\sigma$-CFDMs.

In this section, we present some preliminary findings on the relationship between the inductive bias of neural SGMs and that of our proposed $\sigma$-CFDM. We first show that in low-dimensional problems, one may approximate the velocity field of a neural SGM with that of a $\sigma$-CFDM for an appropriate choice of $\sigma$. We then show that this is no longer the case for high-dimensional image datasets such as MNIST with the velocity field $v_t(z)$ parametrized by a Unet. Our strategy will be to extract $k_t(z)$ from a neurally-parametrized $v_t(z)$ and show that in contrast to a $\sigma$-CFDM, the neural $k_t(z)$ does not output convex combinations of training samples for $t \approx 1$.

### H.1    2D datasets

For this experiment, we train a simple 3-layer MLP on the flow-matching objective from (Liu et al., 2023), whose theoretical optimum is attained by the velocity field $v_t(z) := \frac{1}{t}(z + (1-t)\nabla \log \rho_t^*(z)$. (Recall that for all $0 \le t \le 1$, $\rho_t^*$ is a mixture of Gaussians centered at rescaled training samples with common covariance $(1-t)^2 I$.) Our MLP has 64 neurons in each hidden layer and uses Softplus activations. We train this MLP for 20k epochs at a learning rate of $10^{-3}$ using AdamW, and take the target distributions to be the empirical distribution over fixed sets of 500 samples from the 2D "Checkerboard" dataset used throughout this paper and a 2D "spirals" dataset, respectively.

We then sweep over $\sigma \in [0, 2]$ and construct a $\sigma$-CFDM on the same 500 training samples for each value of $\sigma$. For each value of $\sigma$, we then sweep over $t \in (0, 1)$, draw batches of samples from $\rho_t^*$, and compute the average squared $L_2$ distance between the neural velocity field $v_t$ and the velocity field $v_{\sigma,t}$ defined in equation 6. We

report the average squared $L_2$ distance across $t$ for each value of $\sigma$ in Figure 14. The unsmoothed CFDM's velocity field $v_t^*$ is a poor approximation to the neural SGM's velocity field, indicating that the neural SGM does not learn the closed-form score for its training set. However, smoothing the CFDM significantly improves the quality of the approximation, with $\sigma = 0.45$ achieving an 88.8% reduction in squared $L_2$ error against the neural velocity field relative to the unsmoothed velocity field on the checkerboard dataset, and $\sigma = 0.15$ achieving an 81.1% reduction in squared $L_2$ error on the spirals dataset.

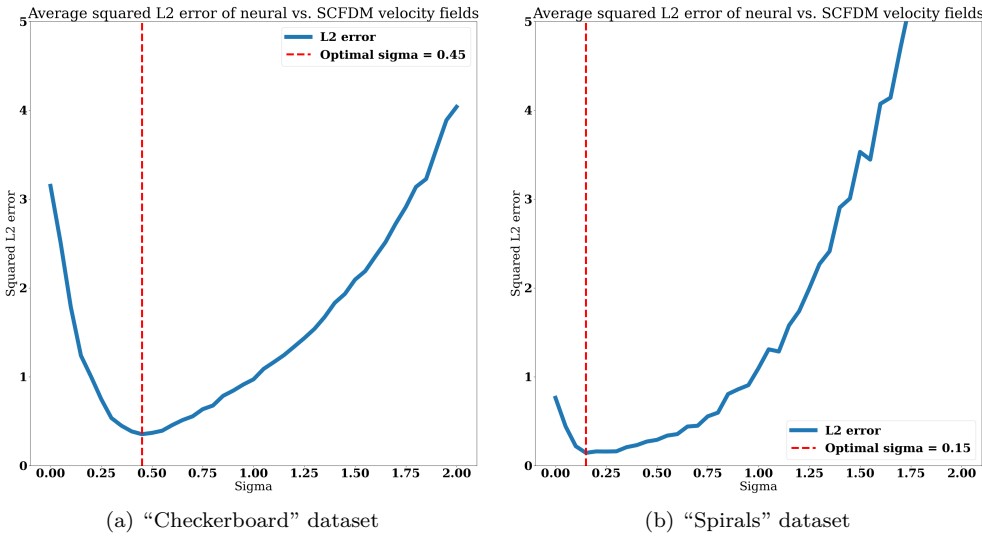

(a) "Checkerboard" dataset          (b) "Spirals" dataset

Figure 14: An unsmoothed CFDM's velocity field is a poor approximation to a neural SGM trained on the same dataset, but smoothing it significantly improves the quality of the approximation.

In Figure 15, we compare samples drawn from the neural SGM (in red) to samples drawn from a $\sigma$-CFDM with the optimal values of $\sigma = 0.45$ and $\sigma = 0.15$ for the checkerboard and spirals datasets, respectively. While each distribution's samples are qualitatively similar, we note in particular that the $\sigma$-CFDM places less mass near extreme points of the support of the training data. We observed a similar phenomenon in Appendix D and conjectured that it results from $\frac{1}{M}\sum_{m=1}^{M}k_t(x + \sigma\epsilon_m)$ converging towards a deterministic quantity for sufficiently large values of $M$. We further conjecture that this phenomenon may partially explain why neural SGMs generalize better on high-dimensional image generation problems, as they place more mass near extreme points of the data distribution and less mass on barycenters of training samples, which typically do not resemble natural images.

Finally, in Figure 16, we compare the velocity fields of a neural SGM and an appropriately-smoothed $\sigma$-CFDM for the checkerboard and spirals datasets at three times: $t \in \{0.1, 0.5, 0.9\}$. We normalize the velocity fields in the top row of each subfigure to facilitate a comparison of the direction of each velocity field, and depict the difference of the non-normalized velocity fields in the bottom rows. Smoothing a CFDM yields a velocity field that accurately approximates the corresponding neural SGM's velocity field for $t$ close to 0, but the accuracy of this approximation deteriorates as $t \to 1$.

## H.2 MNIST

In this section, we will show that if one parametrizes $v_t$ by a Unet and trains it on a high-dimensional image dataset such as MNIST, the resulting model does not behave like a $\sigma$-CFDM. Because sample estimates of the squared $L_2$ distance between the neural $v_t$ and a $\sigma$-CFDM's $v_{\sigma,t}$ are noisy in high dimensions, we do not employ the strategy from the previous section to compare a neural SGM and our $\sigma$-CFDM, but instead study the extent to which a neural SGM points towards convex combinations of training samples in image generation problems.

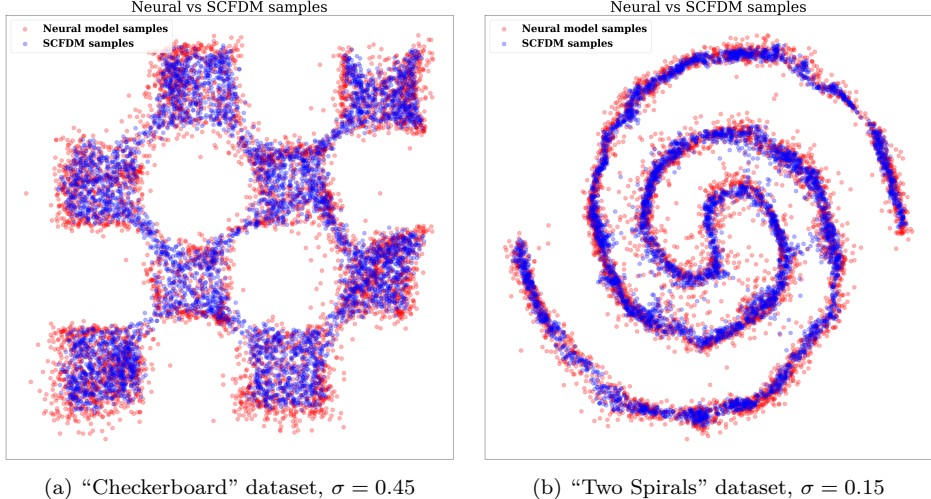

(a) "Checkerboard" dataset, $\sigma = 0.45$        (b) "Two Spirals" dataset, $\sigma = 0.15$

Figure 15: A $\sigma$-CFDM generates samples that are qualitatively similar to a neural SGM on low-dimensional datasets, but sends less mass near extreme points of the data distribution.

Given a velocity field $v_t(z) := \frac{1}{t}(z + (1-t)\nabla \log \rho_t^*(z))$, where $\nabla \log \rho_t^*(z) = \frac{1}{(1-t)^2}(k_t(z) - z)$, one may extract the corresponding $k_t$ by substituting the formula for $\nabla \log \rho_t^*(z)$ into the formula for $v_t$ and rearranging. Equation 4 and Proposition 4.1 show that if $k_{\sigma,t}(z)$ is extracted from either of the unsmoothed or smoothed scores $s_{\sigma,t}$ ($\sigma = 0$ and $\sigma > 0$, resp.), then it must output convex combinations of rescaled training samples for any $z$: $k_{\sigma,t}(z) = \sum_{i=1}^N w_i(z) t x_i$ with $w_i(z) \geq 0$ and $\sum_{i=1}^N w_i(z) = 1$. We will show that for $t \to 1$, the $k_t$ function extracted from a neural SGM's velocity field $v_t$ is *not* well-approximated by convex combinations of rescaled training samples. This will imply that unlike a $\sigma$-CFDM, a neural SGM's score function does not point towards convex combinations of training data.

To this end, we train a Unet on the flow-matching objective from (Liu et al., 2023), whose theoretical optimum is attained by the velocity field $v_t(z) := \frac{1}{t}(z + (1-t)\nabla \log \rho_t^*(z))$. The training set consists of the 60k images of handwritten digits from the MNIST train partition. We train for 10 epochs at a learning rate of $10^{-4}$ using AdamW.

We then draw a batch of 512 samples $x_k$ from the MNIST test partition, fix a value of $t \in (0,1)$, and compute noisy samples of the form $z_{t,k} = t x_k + (1-t)\epsilon$, where $\epsilon \sim \mathcal{N}(0, I)$. We compute the neural SGM's $k_t(z_{t,k})$ for each noisy sample, and use projected gradient descent to regress each $k_t(z_{t,k})$ on the set of rescaled training samples $t x_i$ subject to the constraint that the weights lie in the probability simplex. If the neural SGM is well-approximated by a $\sigma$-CFDM, then we would expect the MSE of this regression to be close to 0.

In the left panel of Figure 17, we depict the average MSE of the regression of each $\frac{1}{t}k_t(z_{t,k})$ onto the convex hull of the training samples $x_i$ as a function of $t$. (We rescale $k_t(z_{t,k})$ by $\frac{1}{t}$ to enable a direct comparison to the convex hull of the training samples $x_i$; otherwise, the MSE values for small $t$ would be small simply because the data has been scaled by $t$.) For small values of $t$, the function $\frac{1}{t}k_t$ extracted from neural SGM is well-approximated by a convex combination of training samples $x_i$, as one would expect for a $\sigma$-CFDM. However, the quality of this approximation deteriorates as $t \to 1$. This indicates that by pointing towards the convex hull of the training data, a neural SGM behaves like a $\sigma$-CFDM for $t \to 0$, but that this behavior vanishes as $t \to 1$.

If a neural SGM fails to behave like a $\sigma$-CFDM for $t \to 1$, then how does it instead behave? The right panel of Figure 17 indicates that for $t \to 1$, the $k_t$ extracted from a neural SGM approximates an optimal denoiser for test data. In particular, by computing $k_t(z_t)$ for a noisy test sample $z_t = tx + (1-t)\epsilon$, one recovers the rescaled test sample $tx$. Because test samples often lie outside the convex hull of a neural SGM's training data, this behavior cannot be replicated by a $\sigma$-CFDM. This provides evidence of a neural SGM's generalization

capabilities by showing that its score points towards regions of the support of the target distribution that are outside the convex hull of the training data.

We further illustrate this phenomenon in Figure 18. When $t = 0.2$, the output of the neural SGM's $k_t(z)$ on noisy data is well-approximated by a convex combination of training samples, as the theory of $\sigma$-CFDMs predicts. In contrast, when $t = 0.8$, a neural SGM's $k_t(z)$ behaves like an optimal denoiser for test data, mapping noisy test samples to their clean counterparts. While a comprehensive study of the generalization of neural SGMs is out of scope for this work, we point to concurrent work such as Kamb & Ganguli (2024), which studies closed-form solutions to the score-matching problem under locality and equivariance constraints to better understand the generalization of SGMs parametrized by convolutional architectures.

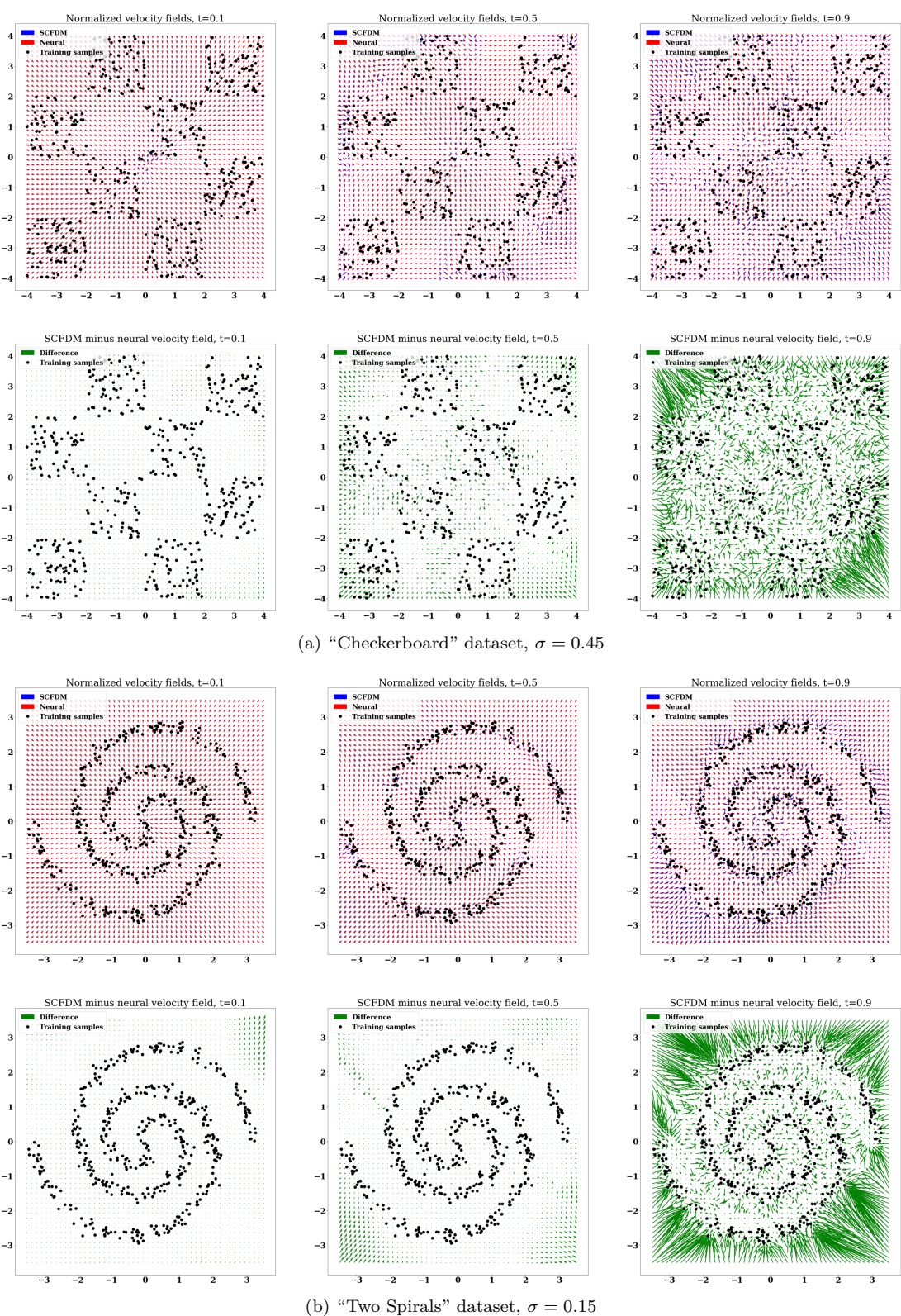

(a) "Checkerboard" dataset, $\sigma = 0.45$

(b) "Two Spirals" dataset, $\sigma = 0.15$

Figure 16: Smoothing a CFDM yields a velocity field that accurately approximates a neural SGM's velocity field for small times, but the accuracy of the approximation deteriorates as $t \to 1$.

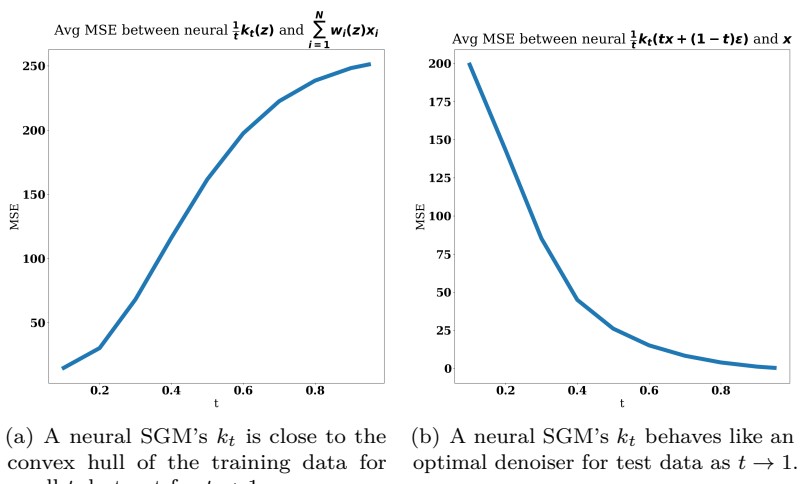

(a) A neural SGM's $k_t$ is close to the convex hull of the training data for small $t$, but not for $t \to 1$.

(b) A neural SGM's $k_t$ behaves like an optimal denoiser for test data as $t \to 1$.

Figure 17: A neural SGM's $k_t$ behaves like a $\sigma$-CFDM's $k_t$ for $t \to 0$. However, as $t \to 1$, it instead behaves like an optimal denoiser for test data.

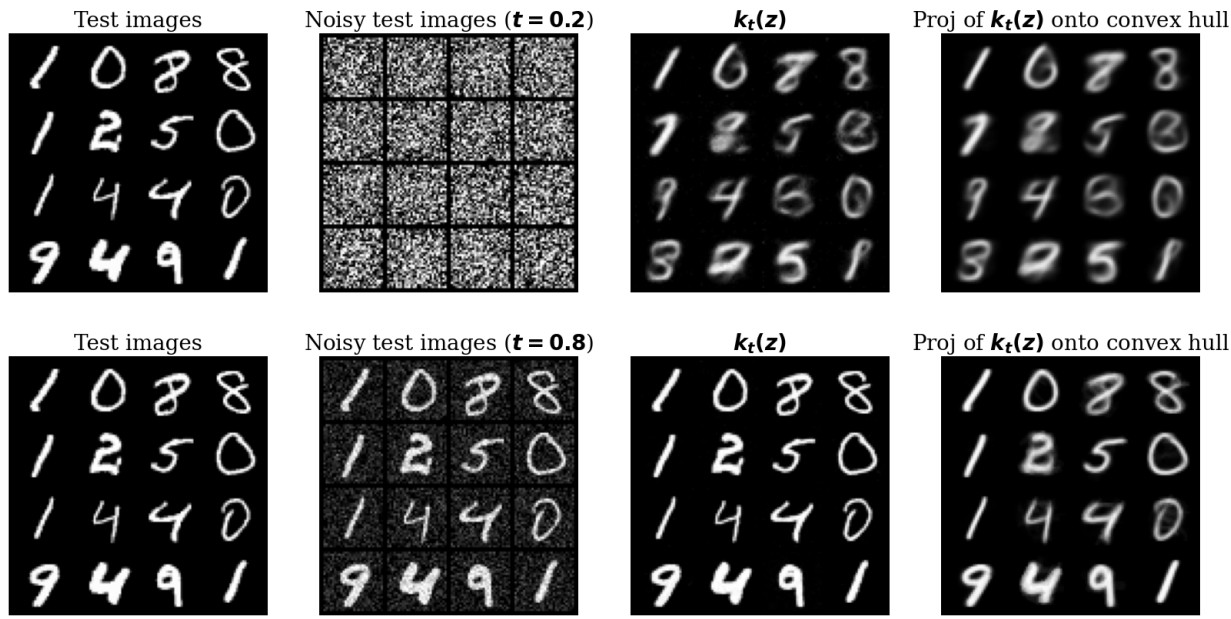

Figure 18: Row 1 shows that for small $t$, a neural SGM's $k_t(z)$ outputs convex combinations of training samples, as one expects for a $\sigma$-CFDM. In contrast, Row 2 shows that for large $t$, a neural SGM's $k_t(z)$ behaves like an optimal denoiser for test data, mapping noisy test samples to their clean counterparts

