# OpenReview forum: "Closed-Form Diffusion Models"
_TMLR — Accepted by TMLR_

### Review · Reviewer_C91g · 2025-03-08

**Summary Of Contributions:**

This paper introduces smoothed closed‐form diffusion models ($\sigma$-CFDMs)—a training‐free approach to score-based generative models (SGMs). In particular, the article studies the impact of sampling with SGMs without training a deep neural network to approximate potentially complex and non-linear score functions $\nabla \log \rho_t$ (following the article notations) associated with a sequence of a perturbed data distributions through Gaussian noise.  While neural approximation methods such as conditional score matching or sliced score matching have demonstrated strong numerical performance, they remain computationally expensive and, to some extent, opaque.

On the contrary, for any finite dataset, the exact score functions can be written in closed form when the training distribution is modeled as an empirical distribution (i.e. a sum of Dirac masses over the training dataset). This approach, called CFDM, provides an exact solution to the so-called, score matching problem.

However, CFDMs lack the ability to promote generalization as $\rho_t$ is merely a mixture of Gaussian random variables centered in the training data point with shrinking variance as $t\to 1$. In the toy example Figure 1 (a) the authors illustrate that the algorithm only outputs copies of the training dataset when sampled with CFDMs. To address this limitation and promote generalization, the authors propose using a smoothed version of the empirical score function rather than the exact empirical score itself.  This is achieved by convolving the score function with centered random noise $\epsilon$, scaled by a parameter $\sigma > 0$ (i.e. in the Gaussian case $\epsilon \sim \mathcal{N}(0,\sigma^2 I_d))$. When, $\rho_t$ is defined as the density of the same flow as in the article, it yields
\begin{align*}
    s_{\sigma,t} (z) & := \mathbb{E} \left[ \nabla \log \rho_t(z + \epsilon) \right] = \frac{1}{(1-t)^2}\left( \mathbb{E} \left[  k_t(z + \epsilon )\right] - z \right) \,
\end{align*}
The remaining expectation over $\epsilon$ is replaced by a Monte Carlo estimator over $M$ samples, yielding
\begin{align*}
    s_{\sigma,t}(z)  & =\frac{1}{(1-t)^2} \left( \frac{1}{M} \sum_{m=1}^M \sum_{i=1}^N \text{softmax} \left( -\frac{\left\| z - tX \right\|^2 -2 \sigma t \langle \epsilon^{(m)}, x_i \rangle }{2(1-t)^2}\right)_i tx_i  - z  \right) \,
\end{align*}

Replacing the score function $\nabla \log \rho_t$ with $s_{\sigma,t}$ and discretizing the associated velocity field yield a generative model called $\sigma$-CFDM. The authors showed that in the infinite time limit the model output will converge towards barycenters of tuples of training points (Theorem 5.1). This could signal better generalization properties as $\sigma$-CFDM sampler ends up interpolating among training points rather than reproducing them one-for-one. However, computing the smoothed score functions $s_{\sigma,t}$ remain costly especially for large dataset. As a solutions the authors proposed to:
- initialize samples directly from the un-smoothed CFDM at some $T>0$ and thus skipping the costlier evaluation of the low-$t$ regime for the smoothed scores.
- replace the full sum over all training points by focusing on nearest neighbors of $z$ plus a small random subsample.

Finally, the authors assess numerically the generalization quality for various $\sigma$, computational trade‑offs associated with the fast sampling schemes and the applicability of $\sigma$-CFDMS to image generation.

**Audience:**

Yes

**Broader Impact Concerns:**

There is not any ethical concerns with the submission.

**Claims And Evidence:**

Yes

**Requested Changes:**

It might be beneficial to clarify, address, or justify the points mentioned in the $\textbf{Weaknesses}$ section above.

**Strengths And Weaknesses:**

$\textbf{Strengths:}$

- The article is well-written and well illustrated relying on both theoretical and numerical arguments. The paper is easy to read and the structure is clear.
- Unlike most SGMs-related works that emphasize on learned approximations of the score function, this work focuses the empirical distribution and the exact closed-form score it implies. This perspective of research is interesting and provides other ways for analyzing memorization vs. generalization in diffusion models.
- The authors acknowledged that their method could scale poorly when the number of training points is large, as the complexity is $\mathcal{O}(N)$, but they provide solution to mitigate this effect. Their numerical experiments are detailed and convincing. Notably, $\sigma$-CFDM is evaluated in terms of sampling quality (Sections 6.3 and 6.4) and compared to DDPM. Additionally, the study includes an analysis of its generalization properties (Section 6.1) and Theorem 5.1.

$\textbf{Weaknesses:}$

- While focusing on ODE sampling can be clearer from an identifiability standpoint, SDE-based approaches often yield better sample quality in practice. Also I would be curious to know if you tried it and if you have reason to believe whether, the continuous noise, inherent to the discretization scheme of SDE such as Euler-Maruyama or Exponential Integrator schemes could work better. It is sometimes said that the noise injection compensate for the error in the discretization scheme on average but maybe they can promote generalization in this closed-form context.
- The paper uses Gumbel noise in Proposition 5.3 to derive an analytical result for the one-step distribution. However, the rationale for noise distributions in actual experiments is unclear—are they Gaussian, Gumbel, or something else? Further discussion about how the choice of noise might impact smoothness or sample diversity would be valuable.
- I believe that the discretization error is not clearly taken into account. For example when the authors say "SGM memorizes its training data and does not generate novel samples" I guess this is theoretical, as real implementations use a numerical solver with a finite step size. This can introduce unavoidable discretization error, which may indirectly help with generalization.
- The paper states that the model samples converge towards barycenters of $M$-tuples of training points, which suggests that the parameter $M$ could be critical in controlling the interpolation behavior between training samples and the degree of generalization. However, I did not find a clear indication of how $M$ is set in the experiments, nor a discussion on its theoretical impact. Perhaps it would be beneficial to see an analysis of the sensitivity of the results to the choice of $M$.

$\textbf{Minor details:}$
- I believe a factor 2 is missing below equation (25) when upper bounding the Lipchitz constant L.
- In equation 23, the variable of integration is missing.
- I believe an exponential is missing in the definition of $\tilde \beta(T)$ in equation (33).
- Does the hypothesis in page 22 : $\left\| z - tx_i\right\| \leq A$ for all $z$ and $tx_i$, imply a restriction to a compact set ?

---

> ### Author Response · Authors · 2025-03-24
> **Response to Reviewer C91g**
>
> **Part 1/2**
>
> Thank you for your thoughtful review and your careful reading of our proofs. We have uploaded a revised manuscript, in which we have addressed many of your comments. We explain these changes and address your remaining concerns below.
>
> *Also I would be curious to know if you tried it and if you have reason to believe whether, the continuous noise, inherent to the discretization scheme of SDE such as Euler-Maruyama or Exponential Integrator schemes could work better. It is sometimes said that the noise injection compensate for the error in the discretization scheme on average but maybe they can promote generalization in this closed-form context.*
>
> We do not expect that adding noise during sampling would promote generalization more usefully than simply controlling $\sigma$. Our proof of Theorem 5.1 shows that in the limit of small step sizes, our model outputs barycenters of training data in the final sampling step. Furthermore, the proof shows that this would be true regardless of the position of the second-to-last sample – it would simply change which barycenter is picked out. Consequently, adding zero-mean noise to the velocity field during sampling would still result in our model outputting barycenters of training samples, provided the noise variance vanishes as $t \rightarrow 1$. (If there were noise in the final iteration, then the model would output noisy barycenters of training samples.)  Similarly, adding noise to the velocity field of an unsmoothed CFDM does not induce generalization: Such a model also outputs training samples in the limit of small step sizes if the noise variance vanishes as $t \rightarrow 1$.
>
> We expand on this discussion in Appendix F of the revised manuscript and include an experiment to demonstrate that adding noise to an unsmoothed CFDM’s velocity field does not induce generalization.
>
> *The paper uses Gumbel noise in Proposition 5.3 to derive an analytical result for the one-step distribution. However, the rationale for noise distributions in actual experiments is unclear—are they Gaussian, Gumbel, or something else? Further discussion about how the choice of noise might impact smoothness or sample diversity would be valuable.*
>
> Using Gumbel noise in Proposition 5.3 enables us to analytically derive the one-step sample distribution of a $\sigma$-CFDM by using the Gumbel softmax trick. However, we used Gaussian noise in all of our experiments because it is easy to sample and we did not have reason to believe that varying the noise distribution would materially impact our results, provided one can control its mean and covariance.
>
> In Appendix E of the revised manuscript, we illustrate our model’s robustness to the choice of smoothing noise via an experiment in which we draw CelebA samples in latent space using the same hyperparameters as in Section 6.4, but smooth the CFDM using Gumbel noise with the same mean and variance as the Gaussian noise used in Section 6.4. Smoothing with Gaussian and Gumbel noise results in similar sample quality metrics and qualitatively similar samples.
>
> *I believe that the discretization error is not clearly taken into account. For example when the authors say "SGM memorizes its training data and does not generate novel samples" I guess this is theoretical, as real implementations use a numerical solver with a finite step size. This can introduce unavoidable discretization error, which may indirectly help with generalization.*
>
> Thank you for this interesting point. We have added a new Appendix G to the revised manuscript, in which we first show that an unsmoothed CFDM may in principle output non-trivial barycenters of training samples for large step sizes, but then demonstrate empirically that an unsmoothed CFDM memorizes its training data when sampled with as few as 5 steps. This shows that for step sizes commonly used in practice, one must smooth a CFDM to induce generalization; discretization error alone does not suffice.
>
> *The paper states that the model samples converge towards barycenters of M-tuples of training points, which suggests that the parameter M could be critical in controlling the interpolation behavior between training samples and the degree of generalization. However, I did not find a clear indication of how M is set in the experiments, nor a discussion on its theoretical impact. Perhaps it would be beneficial to see an analysis of the sensitivity of the results to the choice of M.*
>
> Thank you for this suggestion. The parameter $M$ is indeed important for controlling the generalization behavior of our smoothed CFDM, and we empirically study the impact of $M$ on a smoothed CFDM’s samples in Appendix D. In the original manuscript, we studied the impact of $M$ on sampling from low-dimensional datasets. In the revised manuscript, we expand this appendix to also study the impact of $M$ on our latent sampling experiments.
>
> **Part 2/2 follows below**

---

> ### Author Response · Authors · 2025-03-24
> **Response to Reviewer C91g (Part 2/2)**
>
> **Part 2/2**
>
> **Minor details.**
>
> Thank you for your careful review of our proofs. We agree with your comments and have revised the relevant proofs in accordance with them. The revised proof text is highlighted in blue in the updated manuscript. In particular, we agree that the hypothesis that $\|z-tx_i\|\leq A$ for all $z, x_i$ is problematic (because one cannot control the location of $z$ a priori), and have revised our proof of Proposition B.2 to instead control the Lipschitz constant of $k^*_t$ in terms of the diameter $D$ of the set of training points $x_i$. Because a CFDM is constructed from a finite training set, $D$ is always a finite quantity.
>
> **Conclusion.**
>
> Thank you for your review. We hope that we have addressed your comments; otherwise, we would be pleased to make further changes in the camera-ready.

---

### Review · Reviewer_hJJF · 2025-03-09

**Summary Of Contributions:**

This is an interesting paper, attempting to replicate the generalization inducing error of the score in diffusion models by using smoothing methods. This does result in a training-free diffusion model. A simple estimator is proposed and experimental results are provided.

**Audience:**

Yes

**Broader Impact Concerns:**

Not a particular concern.

**Claims And Evidence:**

Yes

**Requested Changes:**

It is clear to me that the introduced method will not fundamentally outperform modern score based models. However, I will put this as a question below to allow answers from authors. Other suggestions will come after that.

1) The samples, say in Figure 7(c), should be compared to properly trained (latent) diffusion models. An interesting question here would be about a fair comparison. How big should the score network of a standard trained SGM be - for a fair comparison to your method? Even if the results are negative (that is SGM will clearly beat your method), include it. See my point below for how to actually change a bit the direction of this paper.

2) The comparison to DDPM is on the butterfly dataset. Why was this not done on faces or something more standard - something that is easier to assess?

3) What would be interesting for me is the following: It is clear that standard SGMs induce an error on the score, compared to the empirical score (which would result in memorizing). However, we don't know either that whether this estimator converges to the ``optimal score''. It would be interesting, even for low-dimensional examples, compute the exact score and trained score and compare the bias in the trained score to your approach. Perhaps some structural information can be used here to adjust your estimator, $\sigma$, or the kernel. This would also contribute to a broader understanding of why these models generalize.

**Strengths And Weaknesses:**

The paper builds on a good (and somewhat well known) observation that the generalization properties of diffusion models must be induced by the score error (as other sources of error are controlled, and without the score error, the model should memorize the data). Based on this, it is a good idea to try to come up with a structured way of perturbing the score to establish generalization.

What I see as a weakness, however, the whole execution of the paper. The authors try to come up with a new idea/new sampling method (fundamentally) by perturbing the score. Obviously, the quality of the samples will not match modern diffusion models. I would rather like to see this idea as an investigation of the inductive bias of the modern diffusion models. See below more on this point.

---

> ### Author Response · Authors · 2025-03-24
> **Response to Reviewer hJJF**
>
> Thank you for your thoughtful review. We have uploaded a revised version of our manuscript and address many of your comments below.
>
> *It is clear to me that the introduced method will not fundamentally outperform modern score based models [...] The samples, say in Figure 7(c), should be compared to properly trained (latent) diffusion models.*
>
> We agree that one should expect neural SGMs to outperform our method in terms of sample quality. Because TMLR’s acceptance criteria are based on alignment between a paper’s claims and evidence rather than SOTA performance, we have sought to avoid making claims suggesting that our method’s sample quality is on par with neural SGMs. To further clarify this point, we have added a discussion of our method’s weaknesses to the conclusion of the revised manuscript, which we highlight in blue. We would be pleased to further clarify our method’s weaknesses anywhere else in the manuscript that you believe may benefit from this.
>
> The purpose of our experiment in Section 6.4 is to demonstrate that our method can yield better-quality samples than a VAE (which is also a training-free sampler given a pre-trained autoencoder) at a marginally higher sampling cost. We believe that a fair comparison to a latent diffusion model on the same dataset would be costly to run and orthogonal to our claims in this section. Please let us know if there is overly-strong language in this section which you feel implies that our method’s sample quality is comparable to that of neural SGMs; we would be happy to adjust any such claims.
>
> *The comparison to DDPM is on the butterfly dataset. Why was this not done on faces or something more standard - something that is easier to assess?*
>
> The purpose of this experiment is two-fold: We wish to show that our method scales computationally to high-dimensional problems, and we wish to show that there exist certain image distributions for which our model’s inductive bias is reasonable. However, we acknowledge that this does not hold for most distributions over natural images, which motivates our latent sampling experiments in Section 6.4. We have adjusted the text in this section to highlight that we do not expect the results in this section to generalize to most real-world image distributions.
>
> *What would be interesting for me is the following: It is clear that standard SGMs induce an error on the score, compared to the empirical score (which would result in memorizing). However, we don't know either that whether this estimator converges to the ``optimal score''. It would be interesting, even for low-dimensional examples, compute the exact score and trained score and compare the bias in the trained score to your approach. Perhaps some structural information can be used here to adjust your estimator \sigma or the kernel. This would also contribute to a broader understanding of why these models generalize.*
>
> We agree that this is an interesting research direction. However, we believe that it is orthogonal to our contributions and would be better-suited for a different paper. For example, “Understanding Hallucinations in Diffusion Models through Mode Interpolation” (Aithal et al.  2024) observes that neural score models learn smooth approximations to the ground truth score, which causes them to “hallucinate” by generating data that lies in between nearby modes of the target distribution. We have added this reference to our related work section and look forward to further research along these lines.
>
> **Conclusion.**
>
> Thank you for your review. We hope that we have addressed your comments; otherwise, we would be pleased to make further changes in the camera-ready.

---

> > ### Comment · Reviewer_hJJF · 2025-03-25
> >
> > Many thanks for your response.
> >
> > Personally I will insist on numerical comparison of your score estimator to a vanilla NN trained score estimator, in terms of its bias. (1) This would be perhaps insightful in a setting where you can also compute the exact score, (2) in a complicated example, trying to focus on differences between a NN learned score and yours, trying to explain in which ways NN learned scores differ, what might be the main reason of their success (and the reason that your method will not outperform them).
> >
> > Otherwise, the paper is weak - proposing a method that will not work that well but hasn't given any insights either as to why. With this structure, I cannot vote accept.

---

> > > ### Author Response · Authors · 2025-03-27
> > > **Response to Reviewer hJJF**
> > >
> > > Thank you for your continued engagement with our work. Please see Appendix H in the latest revision of our manuscript, in which we study the relationship between our $\sigma$-CFDM and neural SGMs on low-dimensional problems (the 2D checkerboard dataset) and on image generation problems (MNIST). We hope that this helps resolve your concerns regarding our manuscript.

---

> > > > ### Comment · Reviewer_hJJF · 2025-03-27
> > > >
> > > > Fantastic many thanks, would you also be able to include, for 2D checkerboard example, the difference between two vector fields closed form score and the learned score? As this example is 2D, one can plot the vector fields with a quiver plot. Difference between two can also be visualised - this will indicate where they differ for 2D checkerboard. It would be interesting to see the same for a 2D Swiss roll data.

---

> > > > > ### Author Response · Authors · 2025-03-28
> > > > > **Response to Reviewer hJJF**
> > > > >
> > > > > Certainly! We have included these plots in the latest revision of our manuscript.

---

### Review · Reviewer_9gc6 · 2025-03-14

**Summary Of Contributions:**

This paper proposes a training-free generation algorithm, namely the smoothed closed-form diffusion models (CFDM). The idea is to utilize the simple closed-form score functions for the empirical distributions in diffusion models, adding noises manually when evaluating the scores, hence guarantee the generation ability of the algorithm. The authors focusing on analyzing the generation ability of the algorithm and implementing the algorithm more efficiently. More specifically, they

1. show the manual noises (M many) encourage the score function to pointing towards the barycenters of M-tuples of training samples. Therefore, the algorithm generates new samples.

2. show the support of the generated samples tends to the barycenters of the M-tuples of the training samples when the step-size tends to zero.

3. apply the nearest-neighbor-based estimator to approximate the score, which save computational cost while preserving the sample accuracy.

Last, they experimentally test their algorithms by checking the generation ability, ablation and computational trade-offs and apply it to high-dimensional image generation tasks.

**Audience:**

Yes

**Claims And Evidence:**

Yes

**Requested Changes:**

1. the notation $x_{i(k,m)}$ for $M$-tuples is a little confusing. Why does the index $i$ mean? A clarification of the notation would be helpful when it is first introduced.

2. what is the trade-off between sample quality and generation ability in smoothed CFMD? How would it depend on parameters $M,\sigma$? Can this trade-off be theoretically quantified (at least for some empirical data distributions)?

3. When we apply the algorithm in practice? How do we choose the optimal $\sigma$? What statistics should we look at when choosing $\sigma$?

**Strengths And Weaknesses:**

**Strengths**

1. The generation ability is an essential in generative modeling. While most of the existing works study the generation ability of the popular diffusion models, the authors bring a new perspective, to find new algorithms with comparable generation ability. In this paper, the smooth CFDM is motivated from properties of diffusion models. But it is much more simpler as it doesn't require to train any neural networks.

2. The authors discuss both the theoretical and experimental properties of the smoothed CFDM, which could benefit broader audiences that are interested in different aspects of generative modeling.

**Weaknesses**

I hope there could be a separate section providing a more complete comparison to diffusion models from both the analytical and empirical perspectives.

---

> ### Author Response · Authors · 2025-03-24
> **Response to Reviewer 9gc6**
>
> Thank you for your thoughtful review. We have uploaded a revised version of our manuscript and address many of your comments below.
>
> *The notation xi(k,m) for M-tuples is a little confusing. Why does the index i mean? A clarification of the notation would be helpful when it is first introduced.*
>
> Thank you for this comment. We have clarified our notation in the revised manuscript and highlighted the new text in Section 4.2 in blue.
>
> *What is the trade-off between sample quality and generation ability in smoothed CFMD? How would it depend on parameters M,σ? Can this trade-off be theoretically quantified (at least for some empirical data distributions)?*
>
> Both $\sigma$ and $M$ are important hyperparameters for controlling the behavior of a smoothed CFDM. We experiment with the impact of $\sigma$ on our model’s generalization in Section 6.1. In the revised manuscript, we have expanded Appendix D to empirically study the impact of M on our model’s samples, both on low-dimensional data and high-dimensional latent sampling. For the sake of completeness, we have also added appendices E, F, G studying the impact of modifying the distribution of smoothing noise, injecting additive noise into the velocity field, and choosing large step sizes, respectively.
>
> *When we apply the algorithm in practice? How do we choose the optimal σ? What statistics should we look at when choosing σ?*
>
> To determine an appropriate value of $\sigma$ in practice, one may perform the experiment analogous to the one in Section 6.1, where one holds out a test set of samples from the ground truth distribution and sweeps over $\sigma$ while measuring the Wasserstein distance between the held-out test samples and the model samples. (For data distributions over images, one may also use a metric such as FID or KID between the model samples and the test samples.) We have clarified this point in Section 6.1 and highlighted the relevant text in blue.
>
> **Conclusion.**
>
> Thank you for your review. We hope that we have addressed your comments; otherwise, we would be pleased to make further changes in the camera-ready.

---

### Decision · Action_Editor_P8fx · 2025-04-23

**Recommendation:** Accept as is

**Comment:**

The discussion with the reviewers led the authors to improve some of the technical claims and, above all, to detail their experimental contributions in various directions by extending the appendices. They also added  some preliminary experimental results comparing the inductive bias of some neural SGMs and of their algorithms. The discussion on the relationship between other SGMs and the proposed approach was a problematic point in the initial version, the revision proposed additional experimental clarifications and has also expanded the various numerical perspectives offered by the methodology.

As stated by the authors, a very active area of research focuses on conditional generation (not only for images). In this perspective, many solutions have been proposed to combine pre-trained score functions with (Sequential or  Markov Chain) Monte Carlo methods to solve inverse problems or conditional sampling problems. A valuable perspective would be to compare the performance of the proposed algorithm, which does not require any training,  with pre-trained scores on posterior sampling tasks.

**Audience:**

Although the proposed method cannot be considered a competitor to the state of the art, it is of interest to the TMLR’s audience.
The paper discusses both the theoretical and experimental properties of the proposed approach which offers various perspectives on  training-free sampling with SGMs which is of interest to the TMLR’s audience.

**Claims And Evidence:**

In this paper, the authors introduce a new sampling algorithm inspired from score-based models that generates new samples without any training procedure. Using the closed-form expression of the score function associated with the empirical distribution provided by the dataset, the authors introduce a smoothed score function to define a new sampler which aims at reaching better generalization properties without training any deep score functions. The smoothed score function depends on a few hyperparameters (a smoothing parameter, and the number of small perturbations to average the weights of the smoothed score).

The authors propose various numerical experiments to illustrate their claims. In particular, they analyze the impact of the smoothing parameter on the generalization capacity with simulated data. They also use their sampler for image generation using the “Smithsonian Butterflies” dataset. This experiment, although simpler than most settings used for state-of-the-art diffusion models, highlight that the approach can scale to moderately high dimensional problems. This claim is also supported by additional experiments  to sample in  the latent space of  pretrained autoencoders on the CelebA dataset.

The paper discusses both the theoretical and experimental properties of the proposed approach and although the authors do not illustrate their claims with state-of-the-art experimental settings, simulated data are used to support the methodology and the image generation experiments display the variety of applications where the method can be used.